# Assessment of Continuous Flow Analysis (CFA) for High-Precision Profiles of Water Isotopes in Snow Cores

Remi Dallmayr[1], Hannah Meyer[2], Vasileios Gkinis[3], Thomas Laepple[5,6], Melanie Behrens[1], Frank Wilhelms[1,4] and Maria Hörhold[1]

[1]Alfred-Wegener-Institut Helmholtz-Zentrum für Polar-und Meeresforschung, Bremerhaven, Am Handelshafen 12, 27570 Bremerhaven, Germany

[2]Institute of Meteorology and Climate Research (IMK-TRO), Department Troposphere Research, Karlsruhe Institute of Technology (KIT), PO 3640, 76021 Karlsruhe, Germany

[3]Niels Bohr Institute Physics of Ice, Climate and Earth, Tagensvej 16, 2200 Copenhagen, Denmark

[4]GZG Abt. Kristallographie, University of Göttingen, Göttingen, Germany

[5]Alfred-Wegener-Institut Helmholtz-Zentrum für Polar-und Meeresforschung, Potsdam, Telegrafenberg A45, 14473 Potsdam, Germany

[6]University of Bremen, MARUM – Center for Marine Environmental Sciences and Faculty of Geosciences, 28334 Bremen, Germany

*Correspondence to*: Remi Dallmayr (remi.dallmayr@awi.de)

**Abstract.** In order to derive climatic information from stable water isotopes of the very recent past, the signal-to-noise ratio in climate reconstructions from ice cores has to be improved. To this end, the understanding on the formation and preservation of the climate signal in stable water isotopes at the surface is required, which in turn requires a substantial amount of snow surface profiles. However, due to its high porosity and poor stability surface snow has been rarely measured, i.e. climate records from firn and ice cores often start at several meter depths and the few discrete samplings of surface snow took large efforts. We here present a new set-up to efficiently measure stable water isotopes in snow profiles utilizing a Continuous Flow Analysis system enabling measuring multiple snow-cores in a reasonable time and with high-quality. The CFA-setup is described and a systematic assessment of the mixing of the isotope signal due to the set-up is conducted. We systematically determine the mixing length at different parts of the system. We measure and analyse six snow cores from Kohnen station, Antarctica, and find the largest contribution to mixing to originate in the percolation of meltwater on top of the melt head. In comparison to discrete measurements, we show, that our CFA system is able to reasonably analyse highly porous snow cores for stable water isotopes. Still we recommend for future developments to improve the melt head with respect to the strong percolation.

## 1 Introduction

Stable water stable isotopes ($\delta^{18}O$ and $\delta D$) in polar ice cores are commonly used to derive paleo-temperatures (Jouzel et al., 1997). In low accumulation areas of the East Antarctic Ice Sheet, the reconstruction of past climates over large time scales, i.e. several interglacial periods are possible (e.g., Petit et al., 1999; Kawamura et al., 2017), while at the same time the reconstructions of shorter time scales, i.e. interannual-to decadal climate variability, is highly problematic (Ekaykin et al.,

2002; Hoshina et al., 2014; Münch and Laepple, 2018). The reason lies in the large stratigraphic noise (Fisher et al., 1985) imposed by (post-) depositional processes such as wind-redistribution, adding non-climatic variability to the local climate record. In fact, ice core records from most areas on the East Antarctic Plateau are dominated by noise (Laepple et al., 2018, Casado et al.,2020). However, recent studies (Münch et al. 2016, 2017) showed that averaging over a large number of independent vertical profiles allows for inferring a common local climate signal from the stacked stable water isotope record. These findings imply, that it needs a high number of high-resolution snow profiles at an ice core drill site in order to quantify the noise. Commonly, snow profiles were sampled manually at a snow pit trench wall or recently by a snow liner technique (Schaller et al., 2016), where snow cores are extracted from the snow pack and are cut manually into discrete samples. However, by increasing the number of necessary snow cores for each site, the work load for the analysis of discrete samples for their isotopic composition is increasing beyond a feasible manner.

To this end the previously applied Continuous Flow Analysis (hereafter CFA) for ice cores can serve as a solution, as ice cores do not have to be cut prior to analysis, but can be melt and analyzed in one piece. By continuously melting a longitudinal section of a core sample on a chemically inert melt head, the CFA provides high-resolution measurements of stable water isotopes at high-pace (Gkinis et al., 2011; Dallmayr et al., 2016; Jones et al., 2017) in parallel with other proxies such as concentrations of chemical impurities (Osterberg et al., 2006; Bigler et al., 2011). However, in the past, firn and ice cores were only analyzed starting at several meter depth. Due to the very high porosity and poor stability of surface snow, the upper meter of the polar snow pack has not been analyzed by CFA, and was only occasionally sampled discretely. One challenge for CFA measurements is the strong percolation taking place in the highly porous snow. Here, we refer to "percolation" as the upward movement of meltwater into the snow due to capillary effects. Percolation in the snow above the melt head leads to mixing of meltwater from adjacent snow layers. This physical mixing inevitably smooths the derived snow record including the isotope signal (Gkinis et al., 2011). A first successful approach to apply CFA to snow-cores is the LISA Box (Kjaer et al., 2021) for measurements in the field, where a snow core of 10 cm-diameter and 1 m-length is melted on a specifically developed melt head. Here the meltwater is analyzed for conductivity and peroxide, allowing a quick estimation of age and accumulation rate with high quality. But due to its application in the field, measurements of other parameters such as stable water isotopes are not possible with this set-up.

In order to enable stable water isotopes measurements in snow cores by CFA, we modified the melting unit of our CFA-system developed at the Alfred-Wegener-Institut, Helmholtz-Zentrum für Polar- und Meeresforschung (hereafter AWI) to use the melt head proposed by Kjaer et al. (2021). We systematically assess the performance and the mixing of the CFA system with respect to the isotope signal by (1) means of isotopic standards, and (2) comparison to discrete measurements. We here refer to mixing as the alteration of the isotope record (originally preserved in the snow) due to the CFA system. We were able to separate and quantify the contribution of different components of the CFA to the overall mixing of the isotope signal, and show that the percolation above the melt head is the major contributor.

## 2. Material and method

### 2.1. Experimental set-up

The system to analyze 1-meter snow cores consists of a melting unit adapted to the geometry of the snow cores, a degassing unit, an electrical conductivity unit, a water isotope measurement unit, a micro-particles detection unit, a fraction collector module, and a datasets synchronization system (Fig. 1). The melt head is constructed such, that it separates the potentially contaminated outer part of the snow core from the clean inner part. The meltwater of the outer part of the snow core is drained to an extra collection unit for non-sensitive measurement, while the meltwater of the inner part is driven to the degassing unit (hereafter Debubbler, Fig. 1) by a peristaltic pump (PP in Fig. 1, Ismatec IPC) through a perfluoroalkoxy tubing with 1/16 " outer diameter and 0.76 mm inner diameter. Downstream of the Debubbler, a second peristaltic pump (Ismatec IPC) drives the now bubble-free water stream to a polyether ketone Manifold (P-150, Idex), from where sub-streams are distributed to the different analytical units through the perfluoroalkoxy tubing of various inner diameters. Via using the Injection Valve (Fig. 1), the analytical units are fed with ultrapure water (Millipore Advantage, Milli-Q $\geq$ 18.2 M$\Omega$.cm-1, hereafter MQ) during periods where no sample is melted.

80

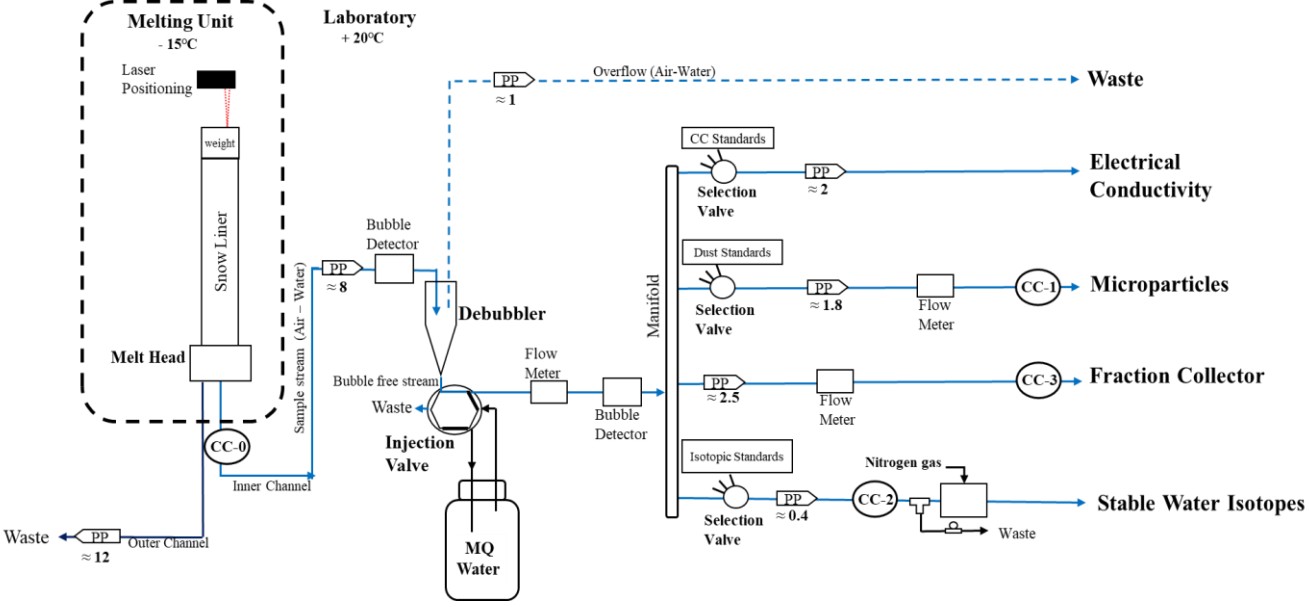

**Figure 1:** Setup of the AWI-CFA system to measure snow cores. The snow-core is held in a snow sample holder and melted on a Melt Head in a -15 °C environment (left hand side). The laser positioning sensor determines the distance to the weight placed on the top of the core. In the laboratory, the conductivity detectors for synchronization (CC-i); an Injection Valve switching between MQ water and snow sample; a degassing unit (Debubbler); a Manifold; Selection Valves for analytical units switching between sample, and standards. The detection of air bubbles takes place upstream and downstream of the Debubbler via using Bubble Detectors. Liquid flow sensors monitor the flow behaviour of the system. Finally, Peristaltic Pumps (PP) carry the streams from the melt head to all analytical units. All indicated flow values are expressed in ml min-1. A detailed scheme of the water isotopes line is given in Fig. A1.

## Melting unit adapted for snow cores

The melting unit for snow cores features a 10 cm inner diameter and 120 cm long tube made of acrylic and positioned centrally above the melt head (Fig. 2a), guiding the sample during the experiment. A light weight (~150 g) is placed atop of the snow core to stabilize the melt-flow, and is covered on both sides with a 1 mm thickness layer of polytetrafluoroethylene to prevent contamination. Following the work of Dallmayr et al. (2016), a high-accuracy laser positioning senor (Way-con, LLD-150-RS232-50-H) determines the distance from the sensor and the top of the weight with a precision of 0.1 mm in order to derive accurate melt-speeds and to assign precise depths to the datasets generated.

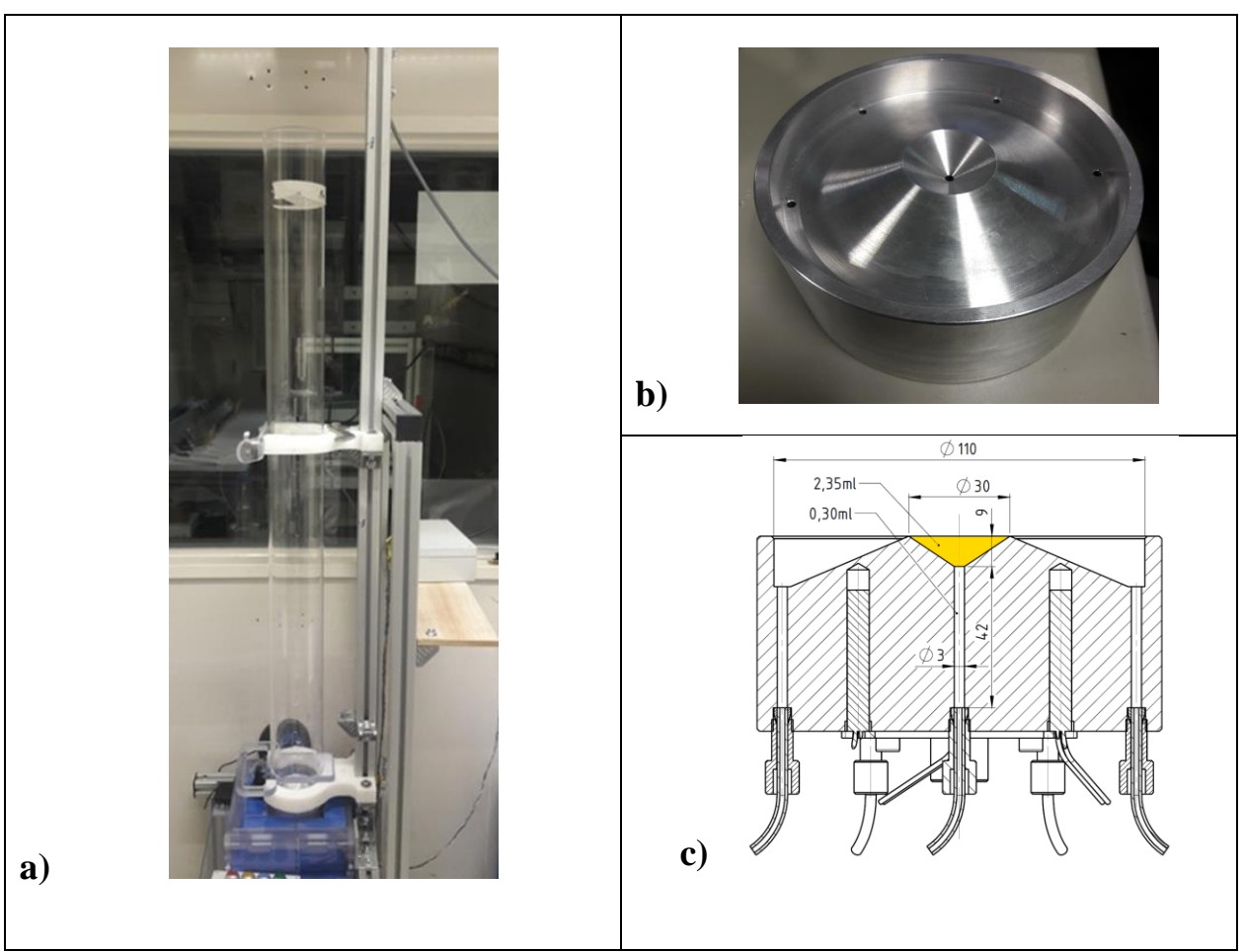

**Figure 2:** Melting Unit. a) Setup of the snow core melting unit. b) picture of the PICE design of the melt head (Kjaer et al., 2021) c) Schematic view of the melt head, showing the dimension of the inner-channel concave volume (orange).

The melt head for snow-cores (Fig. 2b) is based on the Physics of Ice, Climate and Earth, (Copenhagen, Denmark, hereafter PICE) design used by Kjaer et al. (2021), made of aluminum, and manufactured at the AWI in Bremerhaven, Germany. Through a concave volume at the inner-channel (highlighted in Fig. 2c), the shape of the melt head prevents mixing of clean meltwater from the inner-part with the potentially contaminated outer-part of the core sample. The temperature of the melt

head is regulated by a Proportional-Integrated-Derivative (PID) temperature controller (JUMO corporate, Germany) attached to eight 125 W heating cartridges and a conventional thermocouple type J.

**Degassing Unit**

As air in the sample stream leads to significant effects and interferences in the liquid detectors, a separation of air and liquid

via a Debubbler (Fig. 1) is required. The incoming flow drips into a micropipette opened to the air, releasing the air bubbles to the atmosphere by buoyancy and leading to a bubble-free flow downstream. Furthermore, regarding the inhomogeneous distribution of air bubbles in the incoming flow, and unpredictable stops of the core melt, maintaining an amount of water in the Debubbler (safety volume) is essential. To that aim, an overflow-tube (Fig. 1) connected to a peristaltic pump is continuously sucking air, or an overflow if the water-level within the pipette gets in contact. We set the height of this overflow-

tube to a minimal safety volume of ~1 ml. A bubble detector located upstream of the unit (Fig. 1) monitors the variability of air in the incoming stream. A second bubble detector is located downstream of the unit (Fig. 1), monitoring, warning and recording the detection of air.

**Analytical measurements**

The CFA-system performs the online measurement of electrical conductivity (conductivity-cell model 3082, Amber Sciences Inc., USA, Breton et al., 2012) and micro-particles counting and sizing (Abakus, Klotz GmbH, Germany, Ruth et al., 2002). Stable water isotopes ($\delta^{18}O$ and $\delta D$) mixing ratios are continuously measured by a Cavity Ring Down Spectrometer (CRDS, Picarro Inc, USA, Maselli et al., 2013). To obtain the continuity and stability of a micro-flow rate of vapor, as required by the instrument, a stream vaporization module was made based on the original method of Gkinis et al. (2010). A micro-volume tee

is used to split a micro-flow into a 50 μm inner diameter fused-silica capillary from the incoming stream. The waste line featuring a smaller inner diameter, a back-pressure is enabled and pushes the micro-flow through the capillary towards the oven where mixing with dry air occurs before injection to the instrument. To control the back-pressure precisely and efficiently, we divided the waste line using a second tee and added to one sub-waste line a 10-turn micro-metering needle valve (Dallmayr et al., 2016). Schematic and technical details of the water isotope line are provided in Appendix A.

In addition to the online measurements, fractions of the melted sample water are carefully collected under a laminar flow bench for further offline measurements of chemical impurities by Ion Chromatography (normative precision of <10 %, Göktas et al., 2002).

Additional electrical conductivity measurements are performed at the melting unit outlet (CC-0 in Fig. 1, Amber Sciences model 1056, USA) as well as near the inlet of each detection unit (CC-1 to -3 in Fig. 1, contactless conductivity measurement,

Edaq, Australia). Such duplicated measurements allow for a straightforward and efficient synchronization of the different datasets during data processing (Dallmayr et al., 2016).

**Control, data acquisition and processing.**

All devices are connected to the controlling computer, using a software developed with LabVIEW 2012. Drivers are either
provided by manufacturers (pumps, flowmeters) or developed to suit the purpose (Laser positioning, actuated valves, bubble detectors, all analytical units). Analytical data are recorded every second, and are processed after the experiment by using a self-made code developed with LabVIEW 2012.

**2.2 Assessment of mixing**

The determination and calculation of the mixing of the stable water isotope signal was realized using algorithms developed with the software R (R Core Team, 2018).

**Characterization of mixing using step functions**

CFA-systems are known to diffuse, mix and attenuate the original isotope signal (Gkinis et al., 2011; Jones et al., 2017). The
resulting smoothing of the original signal can be described as a mathematical convolution:

$$\delta_m(t) = [\delta_0 \circledast G](t) = \int \delta_0(\tau)G(t-\tau)d\tau \tag{1}$$

with $\delta_m$ the measured value and $\delta_0$ the original (isotopic) value of the sample at time $t$. G is a smoothing filter denoting the impulse response of the system and $\circledast$ refers to the convolution operation.

We here address the characterization of this mixing by analyzing the impulse response of the system (Fig. 4), i.e. we analyze
the derivative of the response of the system to an instantaneous (step) isotopic change. Previously two approaches have been proposed to treat the step response:

First, Gkinis et al. (2011) fit this so-called step response of the system to a scaled cumulative distribution function (hereafter CDF) of a normal distribution, as:

$$\delta_{normal}(t) = \frac{A}{2}\left[1 + erf\left(\frac{t-t_0}{\sigma\sqrt{2}}\right)\right] + B \tag{2}$$

with A and B the isotopic values of the step scaled, $t_0$ the initial time, and $\sigma$ the standard deviation. All parameters are determined by means of least square optimization. In the case of a normal distribution, the impulse response of the CFA-system is described by a Gaussian impulse probability density function (hereafter PDF):

$$G_{normal}(t) = \frac{1}{\sigma_{normal}\sqrt{2\pi}}e^{-\frac{(t-t_0)^2}{2\sigma_{normal}^2}} \tag{3}$$

Here, the standard deviation of the Gaussian PDF ($\sigma_{normal}$) characterizes the mixing length of the system, expressed in seconds.
Later, these values can be converted into a mixing length expressed in mm by applying the measured melt-speed.

Second, because of the skewed shape of the impulse response, Jones at al. (2017) proposed an implementation from normal CDF to two multiplied lognormal CDFs ($\delta_{log-log}(t) = \frac{C}{2}\left[1 + erf\left(\frac{t-t_1}{\sigma_1 * \sqrt{2}}\right)\right]\left[1 + erf\left(\frac{t-t_2}{\sigma_2\sqrt{2}}\right)\right] + D$). This approach provides a slightly better fit to the signal, but the diffusion length is then retrieved using an additional function, fitting the two lognormal CDFs. The derived mixing length requires thus a careful interpretation due to these additional uncertainties.

Our results show very small differences in the resulting mixing lengths obtained by the different approaches (Table 2). We therefore focus this work on the straight-forward Gaussian approach to determine diffusion lengths and assess contributions to the overall mixing.

**Characterization of mixing by comparison to discrete samples**

To evaluate the mixing and the overall performances of the CFA-system, we compare the results obtained from CFA with discretely measured samples from the same snow cores. We use six 1-meter long snow cores (with the names KF13 – to KF 18), originating from Kohnen station (0°04E 75°00 S, 2892 m.a.s.l.), Dronning Maud Land (Oerter et al., 2009). This area is characterized by an annual mean temperature of -43°C (Weinhart et al., 2020) and an accumulation rate of 75 mm w.e. yr[-1] over the last 50 years (Moser et al., 2020). The snow cores were taken following the procedure by Schaller et al, (2016) from

a trench wall excavated during the 2014/2015 (Münch et al., 2017) with a horizontal distance of 5m. The snow cores were stored in carbon tubes, sealed at each end with plastic bags (WhirlPack), and kept inside a Styrofoam box at -25°C. The 1 meter average density of these snow cores ranges between 340 - 345 kg.m[-3] (Münch et al., 2017). High-resolution density measurements display the layered character of the snow, which induces mm-to-cm variations in density and microstructure. However, stable water isotopes do not capture these variations, as the diffusion on site smoothes the signal rather quickly

(Moser et al. 2020).

From all snow cores a longitudinal section (slice) of 25 mm thickness was cut. This slice was further cut into discrete subsamples of 22 mm size in vertical resolution. The discrete samples were analyzed at AWI Potsdam in 2015 (hereafter dataset discrete-15). In 2019, a second discrete dataset was obtained with a similar 22 mm depth-resolution, just prior to the measurement by CFA. For this second discrete dataset (hereafter discrete-19), four cores (KF13-16) were analyzed for isotopic

composition whereas the remaining two cores (KF17 and 18) were analyzed for ion chromatography at AWI Bremerhaven. For the measurement of the isotopic composition of the discrete 2019 data set the instruments Picarro L2120-i and Picarro L2130-i were used. The measurement set-up followed the Van-Geldern protocol (Van Geldern and Barth, 2012). Each sample was injected 4 times. As a measure of accuracy, we calculated the combined standard uncertainty (Magnusson et al., 2017) including the long-term reproducibility and bias of our laboratory by measuring a quality check standard in each measurement

run and including the uncertainty of the certified standards. The combined uncertainty for δ18O is 0.14 ‰ and for δD is 0.8 ‰.

## 3. Results

### 3.1 Assessing the instrumental performance of the analyzer for stable water isotope measurements

In order to describe the stability and performance of the CRDS instrument for continuous analysis of stable water isotopes, a so-called Allan variance test is applied (Allan, 1966) where over a period of more than 12 hours MQ water is continuously injected at the selection valve for water isotopes (Fig. 1). Such a test allows the investigation of the noise and drift of the spectrometer with respect to the integration time.

The Allan variance is defined as:

$$A_{var} = \frac{1}{2(n-1)} \sum_{i=1}^{n-1} (y(\tau)_{i+1} - y(\tau)_i)^2 \tag{4}$$

with $y(\tau)$ the average value of the measurements during an integration interval of length $\tau$, and n being the total number of intervals. The results show the linear decrease of the Allan deviation (square root of Allan variance) up to a minimal deviation for an integration time of ~6000 s (Fig.3). Instrumental drift starts slightly earlier than $10^4$ s (Fig. 3) with low deviation (0.01 ‰ for $\delta^{18}$O and 0.1 ‰ for $\delta$D) up to $4*10^4$ s. As one run (melting 1-meter liner with CFA) takes up to 2000 seconds, our measurement time window of this one meter is well within the non-affected time window for drift. Therefore, one single calibration for each single meter is necessary.

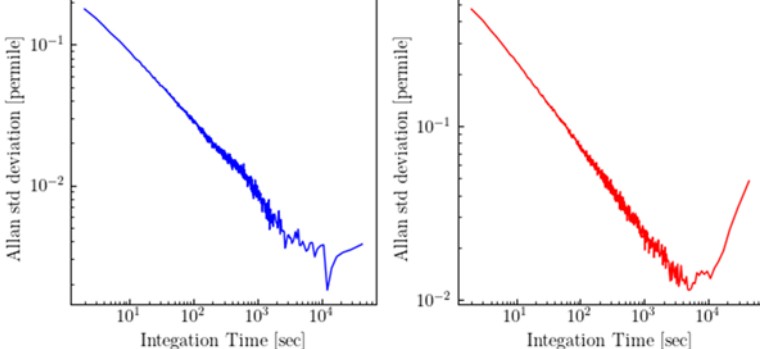

**Figure 3:** >12 h of Allan deviation analysis for $\delta^{18}$O (left panel, in blue), and $\delta$D (right panel, in red).

### 3.2 Calibration to the VSMOW-SLAP scale and deriving the precision of the continuous dataset

Calibrations of the raw data are performed using 3 in-house laboratory standards (Table 1) which are annually calibrated to the international VSMOW-VSLAP scale. Each standard is measured for more than 15 minutes by feeding the CRDS with the standard water via the isotopic selection valve. The values of the last 2 minutes of each run are averaged. The obtained values are plotted against the defined values, the resulting linear regression fitting the 3 points is applied and defines the calibration coefficients (Fig. B1, Appendix B).

|              | NZE            | TD1            | JASE           |
| ------------ | -------------- | -------------- | -------------- |
| $\delta^{18}O$ | -19.85 (0.02)  | -33.85 (0.02)  | -50.22 (0.05)  |
| $\delta D$     | -152.7 (0.3)   | -266.2 (0.3)   | -392.5 (0.4)   |

**Table 1:** Defined Isotopic composition of the in-house laboratory standards used for VSMOW-SLAP calibrations, in ‰. For each standard, the combined uncertainty is given in parenthesis.

The precision of the CFA measurements for stable water isotopes is determined from the standard deviation (1 SD) of the last 2 minutes of each injected standard run. The derived precision from 18 calibration runs (N=18) is 0.24 ± 0.02 ‰ and 0.47 ± 0.04 ‰ for $\delta^{18}O$ and $\delta D$, respectively.

### 3.3. Mixing length derived from step function tests

We estimate the mixing length using isotopic liquid standards across different experiments (details given in Table C1 in the supplement). First, during each calibration procedure, three abrupt isotopic changes are applied (dataset CRDS-line, Fig. 4) at the water isotopes selection valve (Fig. 1). These steps range between ~100 ‰ and ~140 ‰ (Appendix C), and show no dependency between isotopic step size and diffusion length. In a second experiment, we applied an isotopic step (~230 ‰) at the melt head with its concave volume filled (dataset MH-filled, Fig. 4). Finally, the same step was then applied with the
concave volume empty (dataset MH-empty, Fig. 4).

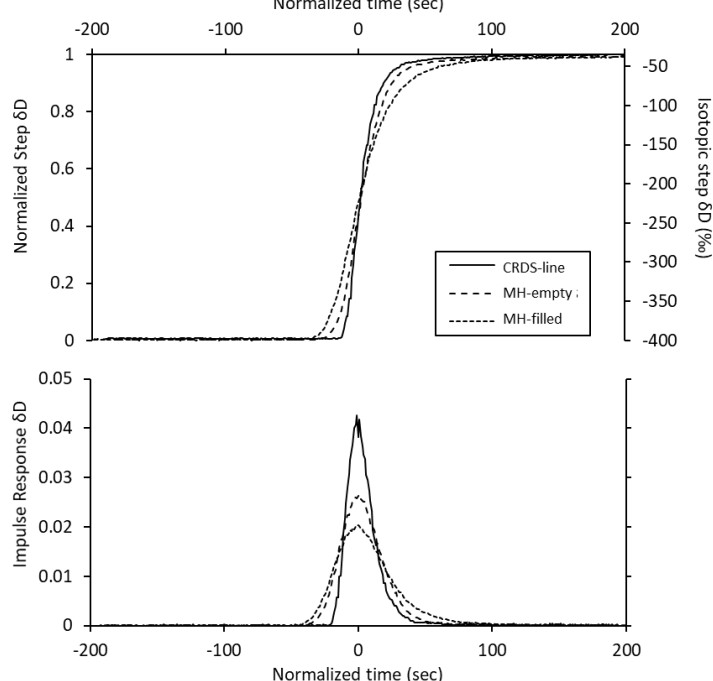

**Figure 4:** Isotopic step and impulse responses obtained for the three experiments realized. Upper panel: The different normalized step functions. Lower panel: the corresponding impulse responses.

Using equations (1) to (3), we compute the mixing lengths *(σ)* of our set of experiments ($\sigma_{MH\text{-}filled}$, $\sigma_{MH\text{-}empty}$, $\sigma_{CRDS\text{-}line}$). The results for both isotopologues $\delta$D and $\delta$18O are very similar (Appendix C, Table C2), consistently showing a slightly longer diffusion length for $\delta$D. Therefore, we focus our study on the results for $\delta$D (Table 2). Furthermore, in addition to evaluating the mixing length induced by the whole CFA system, the combination of the three datasets allows us to distinguish between the contributions of the different parts of the system. Assuming independent mixings from each other from the melt-head (MH) to the water isotopes selection valve (WI-SV), and later to the CRDS instrument, the total mixing filter is the sum of the variances of each mixing filter along the CFA-system.

$$\sigma^2_{CFA-system} = \sigma^2_{MH-filled} = \sigma^2_{MH} + \sigma^2_{MH\ to\ WI-SV} + \sigma^2_{CRDS-line} \tag{6}$$

We can evaluate by quadrature difference (1) the mixing length induced by the concave volume of the melt head ($\sigma_{MH} = \sqrt{\sigma^2_{MH-filled} - \sigma^2_{MH-empty}}$), as well as (2) the mixing length induced downstream of the melt head to the isotopic selection valve ($\sigma_{MH\ to\ WI-SV} = \sqrt{\sigma^2_{MH-empty} - \sigma^2_{CRDS-line}}$).


| | $\sigma_{MH\text{-}filled}$ | $\sigma_{MH\text{-}empty}$ | $\sigma_{CRDS\text{-}line}$ | $\sigma_{MH}$ | $\sigma_{MH\ to\ WI\text{-}SV}$ |
|---|---|---|---|---|---|
| Skew | *20.0 (1.9)* | *13.7 (1.1)* | *11.5 (2.2)* | | |
| | **12.7 (1.2)** | **8.7 (0.7)** | **7.3 (1.4)** | **9.2** | **4.7** |
| Normal | *21.6 (2.4)* | *14.5 (1.2)* | *12.6 (1.8)* | | |
| | **13.6 (1.5)** | **9.2 (0.7)** | **8.0 (1.1)** | **10.0** | **4.5** |

Table 2: $\delta$D mean mixing lengths derived from the skew and normal PDFs, expressed in seconds (italic) and in mm (bold). Values in parenthesis represent 1SD. The conversion of seconds to mm is based on a melt-speed of 38 mm.min$^{-1}$. $\sigma_{MH}$ and $\sigma_{MH\ to\ WI\text{-}SV}$ correspond to differences in quadrature between $\sigma_{MH\text{-}filled}$ and $\sigma_{MH\text{-}empty}$, and between $\sigma_{MH\text{-}empty}$ and $\sigma_{CRDS\text{-}line}$, respectively.

We find a total mixing length of the AWI CFA system of ~14 mm (Table 2), indicating that the majority of the instrumental mixing is induced by the concave volume of the PICE design melt head (10 mm mixing length), closely followed by the CRDS-line (8 mm). The tubular section in between, composed of tubes and the ~1 ml safety volume of the debubbler, shows a significantly smaller contribution (4.5 mm).

## 3.4. Snow-core continuous records versus discrete records

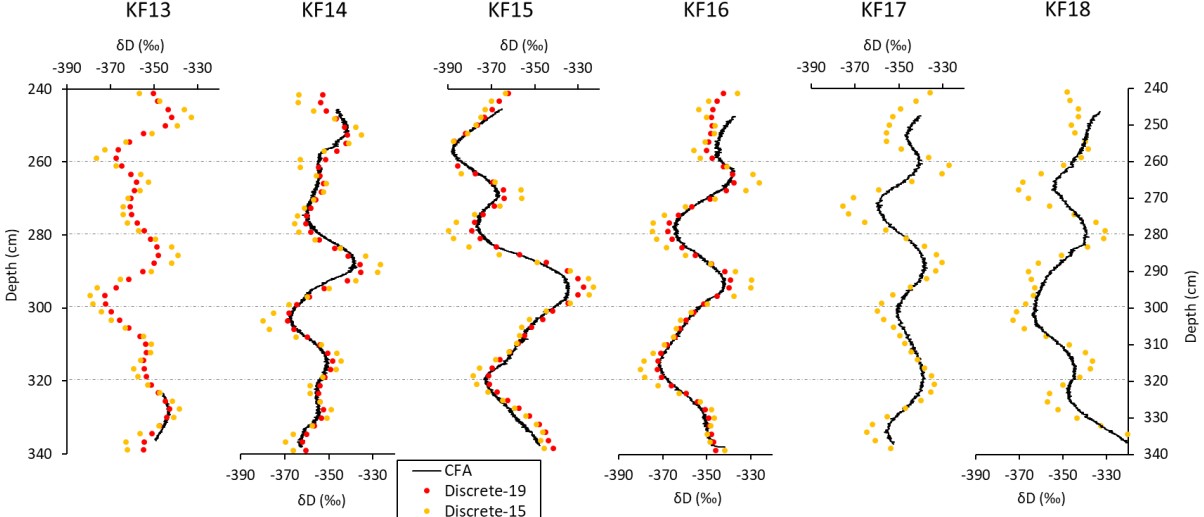

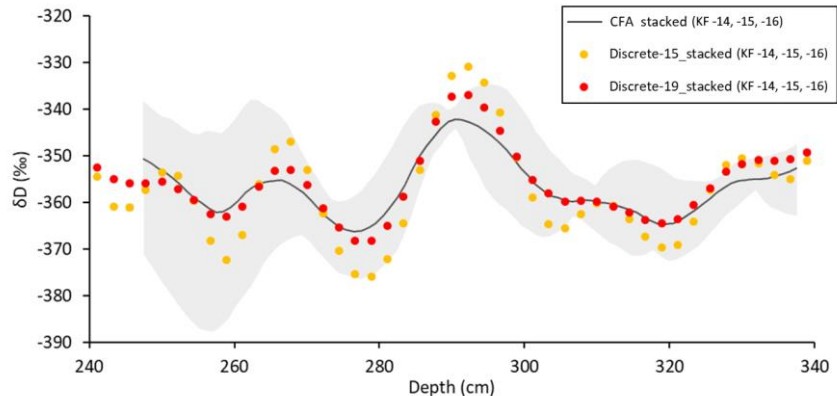

**Figure 5: Isotopes profiles of the 6 snow cores.** Upper panel: 240-340 cm depth δD profiles of the six cores, 5 meters spaced. Datasets of continuous measurement (black lines), discrete-15 (orange markers) and discrete-19 (red markers) are shown. Note the small 150 mm portion of reliable continuous analysis for the core KF13, due to issues during the run. Because of the transition from MQ to sample and vice-versa, we removed the top 65 (±10) mm and bottom 20 (±1.3) mm of the continuous datasets KF-14: -18 (details of the exact removed data are given in Table E1, Appendix E). The averaged melt-speed of the 6 runs is 38 mm.min$^{-1}$ (1SD = 9 mm.min$^{-1}$).

Lower panel: Mean datasets of the stacked CFA (black) and discrete (dotted) profiles of the snow cores KF-14, -15, -16. The gray shaded area displays the spread of the three CFA profiles.

## 3.4 Mixing length derived from comparison to discrete measurements

We compare the discrete values from 2015 and 2019 as well as the values obtained from CFA at the single profiles (Fig. 5, upper panel) as well as at the mean profile, averaged over all profiles (Fig. 5, bottom) and quantify the differences. We first observe a mismatch in the depth assignment between discrete and CFA values – i.e., the phase of the records has an off-set of 12.3± 6.4 mm. One reason for the mismatch could be the short-term fluctuation in the melt speed induced by density variations. However, the melt speed is measured with high resolution and variations are included in the depth assignment. We therefore assume that the depth assignment from the CFA is quite accurate. On the other hand the depth assignment of the discrete samples is less accurate, as the subsampling generates uncertainty in the exact size and position of each sample (i.e. loss of material due to the cutting). This uncertainty in the depth assignment and the corresponding matching to the CFA data may be one major reason for the observed mismatch of discrete and continuous data. Secondly, comparing the amplitude between isotopic minima and maxima show that the continuous record attenuates on average ~17 % of the discrete-19 record, and ~48 % of the discrete-15 record. The strong difference between the two discrete datasets reveal a further and significant diffusional process occurring likely during the storage of the snow cores (Van der Wel et al., 2011). Therefore, we use the discrete-19 dataset to assess the effect of percolation, while the off-set between the two discrete datasets is discussed later as an indicator of diffusion during storage (discussion).

**Separating the effect of percolation from the overall mixing**

When continuously analyzing a snow-core, due to the high-porosity of the upper-meters samples, capillary action (Colbeck, 1974) forces the melted water at the MH to lift upwards, enabling percolation (Fig. 6). Thus, by comparing the continuous
dataset with a percolation-free discrete dataset, we aim to assess the mixing induced by the percolation.

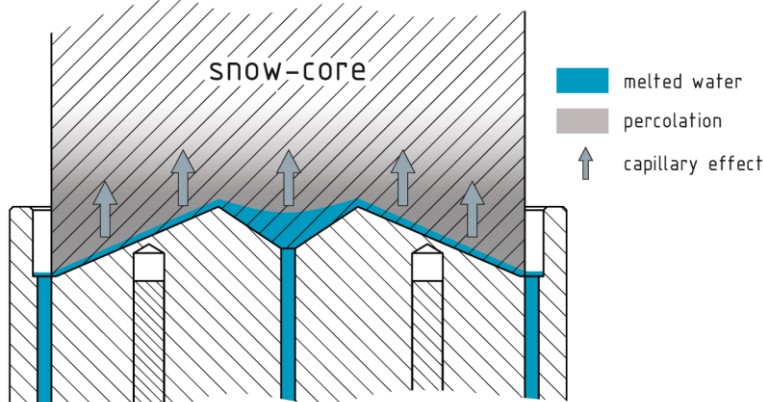

**Figure 6:** Illustration of the melting experiment showing the percolation induced by the low density of the sample combined to the melt-water reservoir within the inner concave volume of the melt head.

In order to retrieve the mixing length induced by the continuous analysis of snow-cores with respect to the discrete dataset, we convolved the discrete signal with a family of impulse responses of different mixing lengths. A set of discrete-convolved signals of flattened extrema are obtained, and each signal is compared to the CFA signal (previously smoothed on 22 mm to ensure the same averaging than the discrete samples). Then, the minimum of root-mean-square (RMS =

$\sqrt{mean\left((x_i - x_j)^2\right)}$) between convolved signals and the continuous record allows for the identification of the adequate
mixing length $\sigma_{CFA\text{-}discrete}$.

| | KF13 | KF14 | KF15 | KF16 | KF17 | KF18 |
|---|---|---|---|---|---|---|
| $\sigma_{CFA-discrete}19$ Normal-CDF | x | 29 | 32 | 31 | x | x |
| $\sigma_{CFA-discrete}15$ Normal-CDF | x | 49 | 52 | 57 | 56 | 57 |

**Table 3:** Mixing lengths (in mm) of the continuous profiles of $\delta$D, as related to the discrete-19 dataset and to the discrete-15 dataset. No results are available for the core KF13 due to the too short continuous dataset.

We find average mixing lengths of approximately 30 mm for the discrete 2019 dataset, and 54 mm for the discrete 2015 dataset (Table 3). Following equation (6), we separate the mixing from the percolation at the melt head from the remaining CFA system by:

$$\sigma_{CFA-discrete19}^2 = \sigma_{CFA-system}^2 + \sigma_{percolation}^2 \tag{7}$$

$\sigma_{CFA\text{-}discrete19}$ refers to the overall mixing retrieved from the convolution of the discrete 2019 signal. $\sigma_{CFA\text{-}system}$ is the observed mixing of 14mm of the CFA system derived from experiments with the step function (section 3.3).

By retrieving the quadratic difference of CFA-discrete19 – CFA system, we can compute the mixing length induced by the percolation at the melt head with ~27 mm. This length is twice the length induced by the experimental system. Thus, percolation is the limiting factor on retrieved signal resolution when applying CFA on snow cores.

## 4. Discussion

The need to address local spatial noise in the stable water isotope ice core records from low-accumulation rate areas such as Kohnen station, Dronning Maud Land, with an accumulation rate of 200-300 mm recent annual gain of snow (Münch et al., 2017) motivates to analyze low-density snow cores with less efforts compared to discrete sampling. In order to assess the ability of a CFA system to analyze these very low density (e.g. (<400 kg.m$^{-3}$, Laepple et al., 2016) cores we here used an adapted melting unit of the CFA system to anaylze in total 6 snow cores from Kohnen station. We find a good agreement between the different isotope profiles of the different snow cores. The derived mixing length of the CFA system with respect to the isotopic signal is found to be similar to the improved CFA-CRDS system at University of Colorado (mixing length for $\delta$D of 21.6 seconds, section 3.3 of this work, versus 19.1 seconds, Jones et al., 2017). These agreements in derived mixing lengths indicate, together with our quantitative estimates (Table 2), that a major contribution to the mixing is inherent in the usage of the Picarro instrument (CRSD line).

### 4.1 Suggestion to address the percolation

The largest contribution originates from the percolation at and above the melt head. We assign the percolation to the design of the melt head with and its large inner concave volume (Fig. 2). To overcome the high-level of mixing induced by the percolation, it is crucial to prevent the formation of a reservoir of melt-water in contact with the snow-core, specifically in the inner volume at the melt head. Thus, a new design of inner channel offers a possibility towards this aim. A flat surface covered with boreholes will allow for an efficient and uniform evacuation of the melt-water, limiting the suction upward. In addition to addressing the percolation, such design will significantly increase the experimental performances (section 3.3) in order to ultimately offer the necessary quality to resolve reliably the full isotopic cyclicity in the snowpack in low accumulation rate areas.

The significant contribution of the Picarro analyzer to the mixing length of the CFA system (CRDS-line, section 3.3) could be addressed as well, and requires likely a collaboration with the manufacturer to improve the analytical unit itself (e.g. reduction of the large volume at its inlet before a pressure-drop to the 40 Torr cavity).

## 4.2 The effect of variations of melt speed on the mixing length

Further we explored the sensitivity of mixing length to melt speed. We melted the snow cores at different melt speeds, varying from 28.7 mm per minute to 49.5 mm per minute (Table 4). As the obtained mixing lengths do not show any trend (see Table 3), we are confident that the mixing length is not sensitive to melt speed variations. We however observed variability of melt speed within each core between 10 and 20 % of the mean speed (Table 5), which can be related to small-scale variations of snow density or friction within the sample holder.

|  | KF13 | KF14 | KF15 | KF16 | KF17 | KF18 |
|---|---|---|---|---|---|---|
| Average speed | x | 28.7 | 31 | 33 | 46 | 49.5 |
| Standard deviation | x | 3.8 (13 %) | 4.4 (14 %) | 5.6 (17 %) | 9 (19 %) | 5.3 (11 %) |

**Table 4:** Melt speed statistics for the different snow cores, expressed in mm.min$^{-1}$. The upper row shows the averaged melt speed of each run, while the lower row indicates the corresponding variability (standard deviation observed, in mm.min$^{-1}$), and in parenthesis as a percentage of the mean value.

## 355   4.3 Spatial variability

The six firn cores were taken in the vicinity of Kohnen station with a distance of 5m. Their different profiles display stratigraphic noise, i.e. spatial variability. The variability in the isotope profiles due to this spatial variability is in the range or larger (Figure 5, lower panel) than the error obtained due to the mixing by the CFA, i.e. the reduced amplitudes compared to discrete data or delay in the signal compared to the discrete data. This strengthens the potential in using CFA to analyze many

snow cores of one site to generate an average isotope profile (Münch et al, 2017).

## 4.4 Diffusion of the isotope signal during storage of snow cores

From comparing the CFA data with discrete measurements (Fig. 5), we observed a significant difference between the stable water isotopic profiles of the same snow cores sampled at different times.

This difference is not related to instrumental induced mixing, but instead indicates the effect of long-term storage of snow samples. We assume that diffusion, known to occur in snow and firn (Gkinis et al., 2014) also occurs in the cold storage environment. Using our results, we can now quantity this storage-induced diffusion. Assuming that the mixing of the discrete-15 dataset corresponds to the mixing of the discrete-19 dataset convolved with an independent smoothing filter induced by the storage, comes:

$$\sigma^2_{CFA-discrete15} = \sigma^2_{CFA-discrete19} + \sigma^2_{storage\,15-19} \qquad\qquad (8)$$

Using the calculated mean mixing lengths $\sigma_{CFA\text{-}discrete19}$ = 30 mm and $\sigma_{CFA\text{-}discrete15}$ = 54 mm (Table 3), we derive a diffusion length of approximately 45 mm. These findings indicate that during the 4 years of storage (from the first analysis in 2015 to the second analysis in 2019), the isotope signal in the snow cores was smoothed by this diffusion length.

Additionally, we computed for each 1-meter long snow core the mean and variability (standard deviation) of the both discrete datasets (Table 5). The decrease in amplitude indicates an average attenuation of 0.54 ‰ and 4.5 ‰ for $\delta^{18}O$ and $\delta D$, respectively. The mean values show on average an enrichment of isotopic composition of +0.31 ‰ for $\delta^{18}O$ and +1.6 ‰ for $\delta D$, likely due to the repeated contact with laboratory-air when bags are opened, and the loss of sample (frost) in the bag.

| | $\delta^{18}O$ Discrete-15 | $\delta^{18}O$ Discrete-19 | $\delta D$ Discrete-15 | $\delta D$ Discrete-19 |
|---|---|---|---|---|
| KF13 | -45.29 (1.3) | -45.076 (0.85) | -356.31 (11.93) | -355.77 (8.05) |
| KF14 | -45.14 (1.45) | -44.71 (0.85) | -354.85 (12.24) | -353.12 (7.69) |
| KF15 | -46.38 (2.39) | -45.91 (2.05) | -364.36 (19.53) | -361.51 (16.75) |
| KF16 | -44.95 (1.70) | -44.76 (1.16) | -353.87 (14.72) | -353.30 (10.23) |

**Table 5:** $\delta^{18}O$, $\delta D$ means (standard deviation) for each snow core discrete dataset, expressed in ‰.

We show the effect of storage on diffusion lengths for both isotopologues (Fig. D1, Appendix D) based on firn-diffusion model (Gkinis et al., 2014). The model run, assuming a "storage" temperature of -20 ºC, a density of 370 kg.m$^{-3}$, an accumulation rate of 75 mm w.e. yr$^{-1}$ (even though there is no accumulation during storage). For a time window of 4 years, we found a diffusion length for $\delta D$ similar to our observations (Fig. D1). As the diffusivity coefficients are positively correlated to temperature, the diffusion during storage is likely of stronger magnitude than on the East-Antarctic plateau (Fig. D2, Appendix D). However, the strength of the observed diffusion during storage is due to the low density of the snow cores and we do not expect such a strong change for firn and ice core samples from greater depth and with higher density.

## 5. Conclusions

Overcoming the increased stratigraphic noise in low accumulation areas and constraining the isotope signal formation within the upper-firn requires large number of high-quality stacked vertical profiles (Münch et al., 2017). In order to cope with the related challenge of high-pace -quality analysis and paired measurements of various proxies, we present here a CFA-system adapted for analysis of snow-cores using a previously designed melt head for snow cores (Kjaer et al. 2021). Based on standard

step functions and mathematical approaches (Gkinis et al., 2011, Jones et al., 2017), we quantify the mixing of the isotopic signal induced by the CFA system and separate the different contributors. As a result, we identify percolation (especially strong due to the low density) at and above the melt head as a major contributor to the overall mixing of the isotope record. Despite these limitations we show that CFA can be used to obtain reasonable isotope profiles of snow cores, keeping information on both, the spatial and temporal variability. However, for future applications of CFA to snow cores we recommend a new design of the melt head, with a focus on the evacuation of the melt-water. Finally, the isotopic diffusion during storage of snow-core samples requires further investigation, but underlines the need of (1) a strategy to preserve the original record (discrete samples cut in the field) or (2) prompt analysis with techniques such as the CFA.

## APPENDIX A: CFA water isotope line

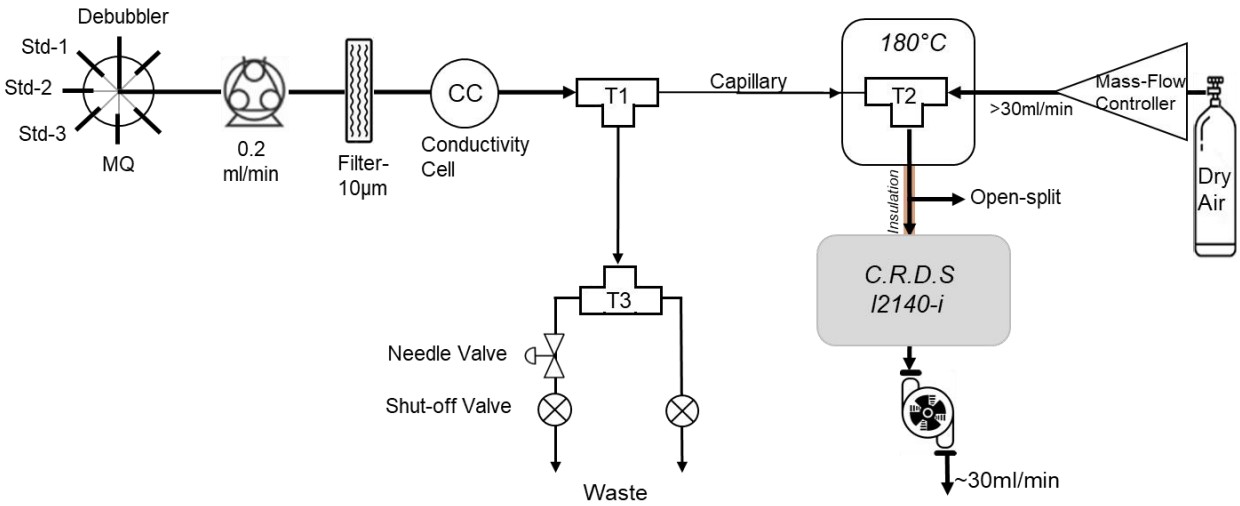

**Figure A1:** Detailed schematic of water isotopes line of the AWI CFA-system.

The selected sample steam is drained at a flow of ~0.2 ml.min$^{-1}$ through PFA tubes of 0.51 mm inner diameter to a 10 μm frits-filter (A-107, Idex), then a synchronization conductivity cell before entering a stainless-steel micro-volume tee (U-428, Idex; T1 in Fig. S1). Here, a micro-flow is split from the incoming stream into a fused-silica capillary tube (50 μm inner diameter),

the rest going into the waste line. Due to the smaller inner diameter of the waste line (0.25 mm), a back-pressure pushes the micro-flow through the capillary towards the oven. To control this back-pressure precisely and efficiently, the waste line is divided using a polyether ketone tee (T3) and added to one of the two sub-waste lines a 10-turn micro-metering needle valve (P-445, Idex).

The sample micro-flow is injected into the stainless-steel tee (T2, Valco ZT1M) mounted in the 180 °C oven, where it vaporizes

instantly and mixes with a controlled flow of dry air (Mass Flow Controller SEC-E40 N2 100SCCM, Horiba company) to form a gas sample with the desired water vapor concentration.

## Appendix-B: 3-points calibration to the VSMOW-SLAP scale and accuracy

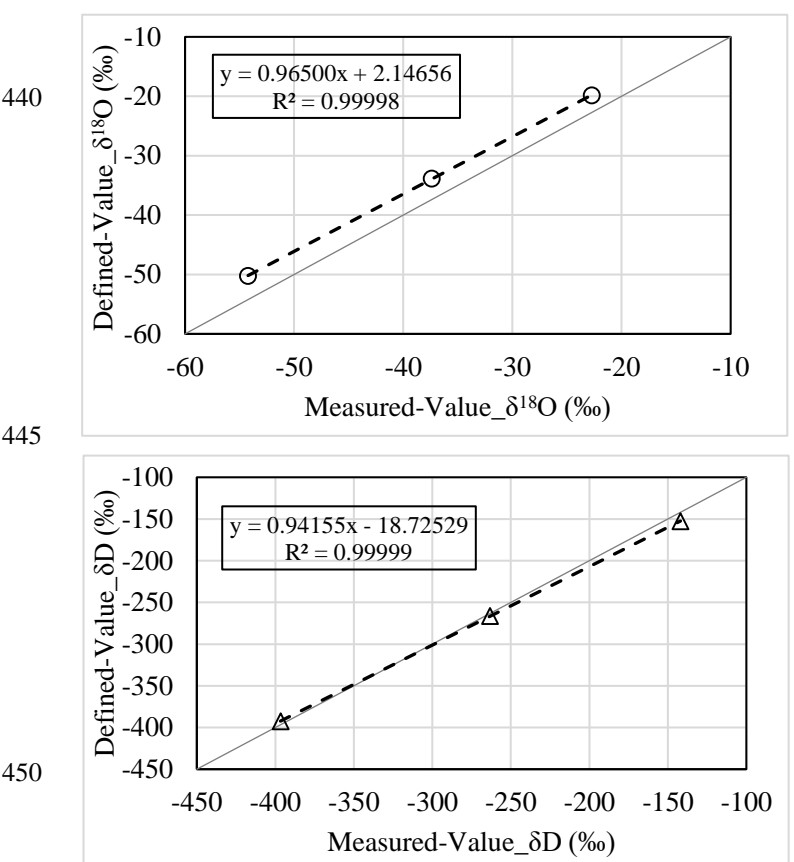

**Figure B1:** VSMOW-SLAP three-points calibration with in-house laboratory standards NZE, TD1, JASE. (Upper panel: $\delta^{18}O$, lower panel: $\delta D$). Thin grey line is the 1:1 line. Datapoints are raw data, i.e. before calibration.

|      | Value after calibration | | Difference with defined value | |
|------|---------------|----------|---------------|----------|
|      | $\delta^{18}O$ | $\delta D$ | $\delta^{18}O$ | $\delta D$ |
| NZE  | -19.804 | -152.422 | -0.045 | -0.278 |
| TD1  | -33.935 | -266.732 | 0.084 | 0.532 |
| JASE | -50.181 | -392.247 | -0.039 | -0.252 |

**Table B1:** Deviations between calibrated and defined values. All values are expressed in ‰.

# Appendix-C: Mixing length derived from step function tests

| Experiment | Abbreviation | Number of runs | Standards switch | Isotopic step size |
|---|---|---|---|---|
| Water Isotopes Calibration (step 1) | CRDS-line | 5 | MQ to NZE | ~100 ‰ |
| Water Isotopes Calibration (step 2) | CRDS-line | 5 | NZE to TD1 | ~120 ‰ |
| Water Isotopes Calibration (step 3) | CRDS-line | 5 | TD1 to JASE | ~140 ‰ |
| Melt head filled | MH-filled | 4 | TD1 to MQ | ~ 230 ‰ |
| Melt head empty | MH-empty | 4 | TD1 to MQ | ~ 230 ‰ |

**Table C1**: Details of the different experiments conducted to assess the overall instrumental mixing, and separating the different contributions along the setup.

| | $\sigma_{MH-filled}$ | | $\sigma_{MH-empty}$ | | $\sigma_{CRDS-line}$ | | $\sigma_{MH}$ | | $\sigma_{MH\ to\ WI-SV}$ | |
|---|---|---|---|---|---|---|---|---|---|---|
| | $\delta^{18}O$ | $\delta D$ | $\delta^{18}O$ | $\delta D$ | $\delta^{18}O$ | $\delta D$ | $\delta^{18}O$ | $\delta D$ | $\delta^{18}O$ | $\delta D$ |
| *Skew* | *19.6 (1.8)* | *20.0 (1.9)* | *13.4 (1.2)* | *11.5 (2.2)* | *10.7 (2.2)* | *13.7 (1.1)* | | | | |
| | **12.4 (1.1)** | **12,7 (1.2)** | **8.5 (0.8)** | **7.3 (1.4)** | **6.8 (1.4)** | **8.7 (0.7)** | **9.0** | **9.2** | **5.1** | **4.7** |
| *Normal* | *20.3 (2.3)* | *21.6 (2.4)* | *13.6 (1.3)* | *14.5 (1.2)* | *11.2 (2.0)* | *12.6 (1.8)* | | | | |
| | **12.9 (1.4)** | **13.6 (1.5)** | **8.6 (0.8)** | **9.2 (0.7)** | **7.1 (1.3)** | **8.0 (1.1)** | **9.6** | **10.0** | **4.9** | **4.5** |

**Table C2**: Means mixing length derived from the skew and normal PDFs, expressed in seconds (italic) and in mm (bold) for both $\delta^{18}O$ and $\delta D$. Values in parenthesis represent 1SD. The conversion of seconds to mm is based on a melt-speed of 38 mm.min-1. $\sigma_{MH}$ and $\sigma_{MH\ to\ WI-SV}$ correspond to differences in quadrature between $\sigma_{MH-filled}$ and $\sigma_{MH-empty}$, and between $\sigma_{MH-empty}$ and $\sigma_{CRDS-line}$, respectively.

## APPENDIX D: Firn diffusion during storage

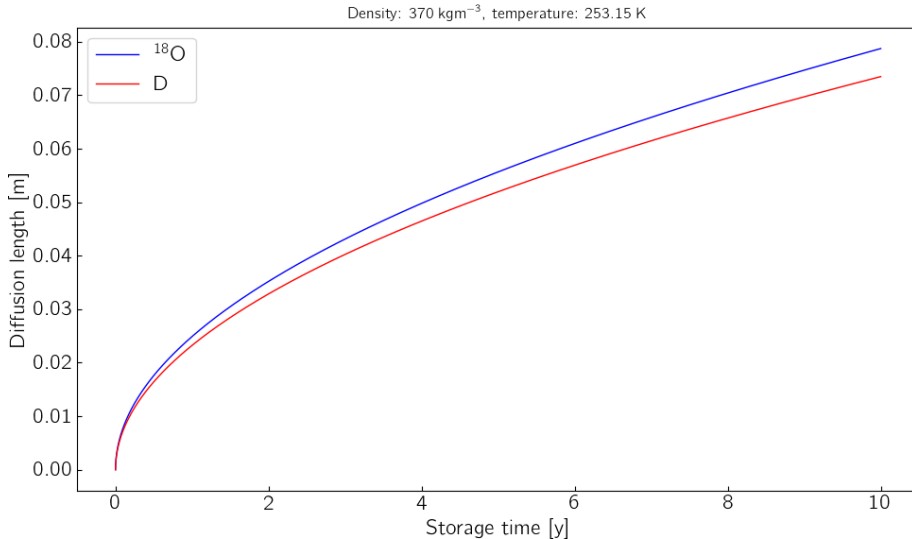

**Figure D1:** Exemplary firn-diffusion length estimates for firn samples with a density of 370 kg.m-3 as function of storage time. The firn diffusion length is computed based on the model by Gkinis et al. (2014), using a temperature of -20⁰C and an accumulation rate of 75 mm w.e. yr-1.

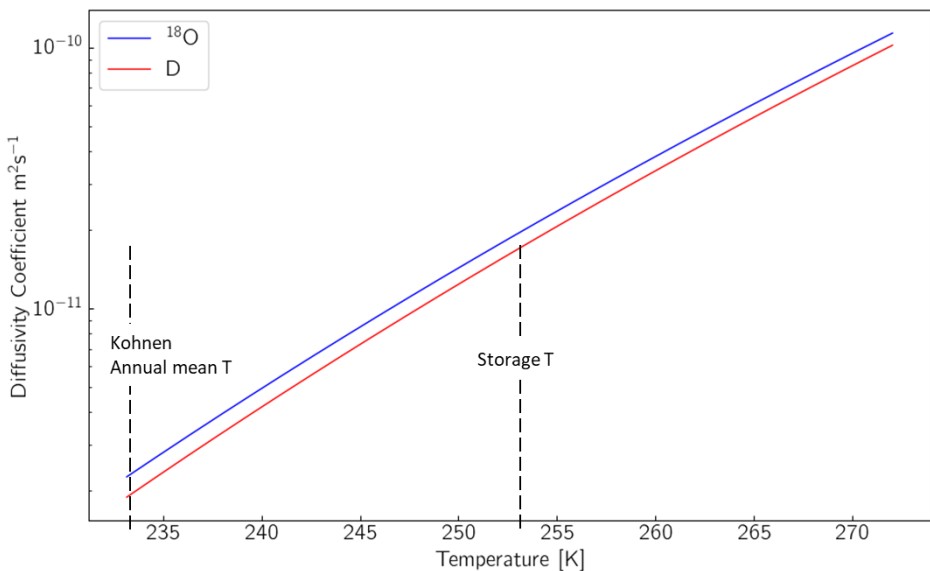

**Figure D2:** Diffusivity coefficients versus temperature, constraints as above, and displaying the annual mean temperature at Kohnen station

(-43⁰C, Weinhart et al., 2020) and the storage temperature (-20⁰C).

## APPENDIX E: Data removed from continuous datasets

|        | Top | Bottom |
|--------|-----|--------|
| KF 13  | x   | 36     |
| KF 14  | 53  | 19     |
| KF 15  | 57  | 20     |
| KF 16  | 66  | 19     |
| KF 17  | 75  | 22     |
| KF 18  | 76  | 21     |

**Table E1:** Data removed from the top (left column), and the bottom (right column) of the continuous datasets due to the transition with MQ water (expressed in mm).

**Authors contribution**

RD developed the CFA-system, designed and proceeded to the tests, experiments, analysis, and co-supervised the diffusion characterization tests. HM took part in all experiments and developed the algorithms to characterize the isotopic diffusion using the R-software. VG provided the routines to calculate the step functions and the results of the model run to describe diffusion during storage. MH developed the research question and supervised the CFA campaigns. Together with TL the scientific background on isotope signal formation in surface snow has been implemented. MB conducted all discrete isotopic measurements and the quality check at AWI-Bremerhaven. All authors contributed to the writing of the manuscript.

**Competing interests**

The authors declare that they have no conflict of interest.

**Data availability**

The algorithms developed in R to characterize the isotopic diffusion are available at: https://github.com/Ice-core-Paleo-Proxies/AWI_CFA_Isotope. The snow-cores discrete samples isotopic datasets 2015 and 2019 are archived at the PANGAEA database, under https://doi.pangaea.de/10.1594/PANGAEA.939208 and https://doi.pangaea.de/10.1594/PANGAEA.969069, respectively.

The snow-cores continuous high-resolution profiles are archived under https://doi.pangaea.de/10.1594/PANGAEA.969073. PANGAEA is hosted by the Alfred Wegener Institute Helmholtz Centre for Polar and Marine Research (AWI), Bremerhaven, and the Center for Marine Environmental Sciences (MARUM), Bremen, Germany.

**Acknowledgmnent**

This project is supported by Helmholtz Research Program Changing Earth – Sustaining our Future. This project was funded by the AWI Strategy fund "COMB-i". TL was supported by the European Research Council (ERC) under the European Union's Horizon 2020 Research and Innovation Programme (grant agreement no. 716092). We are grateful to Helle-Astrid Kjaer and Paul Vallelonga for providing the design of Melt-Head for snow-cores, and the AWI-Workshop for manufacturing. We thank Thaddäus Bluszcz for his help at the first stage of the instrumental development, and York Schlomann for providing all necessary isotopic standards. We further thank Hanno Meyer for the discrete-15 isotopic measurement and Thomas Münch for providing all information for the investigation of the snow cores. Finally, the main author is very grateful to Johannes Freitag and Sepp Kipfstuhl for their relevant inputs on the data analysis, and thank Sonja Wahl and an anonymous reviewer for their helpful comments to improve the quality of the manuscript.

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
