# Peer review of "Assessment of Continuous Flow Analysis (CFA) for High-Precision Profiles of Water Isotopes in Snow Cores"

_EGUsphere, 2024_

## Referee Comment (RC1)

Review for „ Assessment of Continuous Flow Analysis (CFA) for High-Precision Profiles of Water Isotopes in Snow Cores" by Remi Dallmayr et al.

**General comments:**

The authors present an exciting development from the established CFA ice core analysis technique. Routinely deployed to measure stable water isotopes in ice cores, CFA is an indispensable method in ice core science. The here explored possibility of modifying a CFA system to analyze snow cores could facilitate fast, effective and high-resolution generation of recent climate data from snow covered areas and is as such very desirable and of interest to the community. As a supplement, a study result of isotopic diffusion during storage of snow cores is quantified which supports the current understanding of isotope diffusion in firn, as demonstrated by a model comparison.

The authors present an extensive and meticulously generated dataset that is being used to characterize the presented snow-CFA system that combines state-of-the-art system parts from different institutions into an unprecedented measurement instrument. Details and specifications given in the manuscript allow other laboratories to copy the design which is in the interest of open-access science. Diffusion lengths are calculated for individual and combined segments of the CFA system and thus allow to discern and rank the different smoothing-imposing system parts.

The manuscript is presented in an easy-to-follow and concise manner, yet some interesting and relevant details should be explained in more depth which are outlined below. In agreement with a statement in the introduction, the paper concludes that wicking (or percolation) is the main difficulty to overcome when analyzing snow cores with CFA, yet the authors could elaborate further on which system settings are recommended and which limitations remain in terms of density of snow, annual accumulation thresholds etc.

Please elaborate on the following topics:
- Snow property influences: Both water isotope diffusion during storage and percolation are influenced by snow properties such as density and grain size which should be discussed in both sections. Please also add a statement about the density of the snow cores analyzed in this study.
- Melt rate: How was the optimal melt rate chosen and how does this influence the results? Isotope CFA systems are often run in parallel with other instruments and thus the overall CFA melt rate is constrained by more than the water isotope line alone, but if high-resolution water isotope records from snow were the primary goal, which melt rate should be chosen, and what are the constraints? Please also add the statistics of the melt rates of your experiments to the text, both intra and inter snow core variability. Currently, only the mean melt rates are given in the figure captions and are thus hard to find.
- In line 159 the authors declare that they will compare two different diffusion length calculations (normal CDF and two lognormal CDFS) but these results are not presented, although they would be of interest to the reader as the asymmetry in the impulse response is obvious from Figure 5b.
- It would be interesting to see a deconvoluted CFA record in comparison to the discrete samples of 2019 as this would be the final post-processing step for a snow CFA campaign when producing data for climate analysis. Arguably, the reverse was done to find the percolation mixing length contribution, but these

data are not visualized and the statistics (RMS, l. 267) on the agreement with the convoluted discrete samples and the CFA data are not given. As such the reader has no sense for the agreement between discrete and post-processed CFA data. A visualization of such a final data product would help the reader understand the importance of the system characterization and demonstrate the utility of the Snow CFA line.

**Specific comments:**

L. 21: The sentence "With our obtained mixing lengths…" is difficult to understand. Please rephrase.

L. 46: "percolation" is often used in snow science to describe the vertical, gravity-driven water flow in a snowpack. To avoid confusion, please consider exchanging the term "percolation" with "wicking" (as done in (Jones et al., 2017)) or including a short terminology explanation.

L. 46: The authors mention here that percolation is the reason why snow cores have not been measured routinely with CFA. The results presented in this paper support this statement. However, it is not made clear whether the author's conclude that their presented method has overcome this hurdle or which limitations remain with the system presented in this manuscript. I suggest the authors include a "best-practices" or "limitations" section in the discussion.

L. 54: In the introduction it would be helpful to include a short paragraph on the characteristic "mixing of the system" or "smoothing" that all water CFA systems suffer from and that is the drawback of CFA analysis of ice cores. In the current version the "mixing of the system" is first mentioned in L.54 but it is not explained until Section 2.2, (L. 136). Since the "diffusion length" is the focus of the analyses in this manuscript it would help to explain these terms already in the introduction.

L. 59: Is this set-up significantly different from other CFA systems that are being used in ice core analysis laboratories? Explain the differences or cite studies where this set-up has been used previously.

L. 126: What are typical lag times between the different lines?

L. 154: Were all mixing times (in s) converted to mixing lengths (in cm) using the same melt-rate? How stable was the melt-rate of the system during the experiments considering density variability in the snow cores?

L. 166: If available, please add age and density of the analyzed snow cores.

L. 178: Were the Allan variance tests performed by injecting MQ water at the Master IV valve? Would you expect the stability to be different for MQ water injected at the MH?

Fig. 3: I suggest removing the upper row of the figure since both plots don't add much information

Fig 4: Consider moving this figure to the Appendix.

L. 204: In total, 18 calibration runs (or 15? L. 210) were performed over which time span? Was the data calibrated with one averaged calibration function or each experiment dataset calibrated individually?

L. 207: Is the diffusion length dependent on the step size, i.e. the standards used? Please add which standards were used to simulate the step.

L. 208: I would appreciate a table listing the different experiments and the corresponding naming conventions of the different $\sigma$ to ease the reading.

L. 217: Please also give the results for $\delta^{18}O$, even if you focus the discussion on $\delta D$

L. 249: Can this depth assignment mismatch be a result of the system lag between the core hitting the melt head and the time the CRDS is measuring the respective sample? Or is this lag time accounted for?

L. 267: How well do the convoluted and continuous records agree? Please give RMS values.

L. 280: Please give variability of this percolation diffusion length. Discuss the dependence on melt-rate or snow properties (e.g. (Calonne et al., 2012; Yamaguchi et al., 2010; Colbeck, 1974)?

L. 285: Please add short statement on how the snow cores were stored (temperature, sealing, …)

Table 4: Please add the differences between the two measurement campaigns to the table

L. 337: The effect of density or other snow properties on the percolation strength is not discussed up to this point. Please include in discussion.

L. 339: As mentioned above, a clear conclusion and recommendation is missing whether the presented snow-CFA system is recommendable to use and, if not, what restrictions or limitations (accumulation threshold, density etc) apply.

Appendix B: Please add short introduction to the two presented figures and cite the model that is being used. I recommend highlighting AWI storage temperature and Kohnen annual mean temperature to Fig B2.

**Technical corrections:**

L. 21: What does "continuous analyze" mean?

L. 28: delete one "stable"

L. 29: Capitalize "East Antarctic Ice Sheet"

L. 31: change "variabilities" to "variability"

L. 52: Delete "In order"

L. 56: replace "signal, with…" with "signal, and show that…"

Fig. 1: What does "PP" stand for?

L. 80: melting

L. 143: Refer here to Fig. 5

L. 183: replace "optimal" with "minimal"

Fig. 3 and other Figures: Please add panel labels to all plots and refer to them in the captions

Table 1: Please add uncertainties of these in-house standards.

Table 2 caption: Change "means mixing length" to "mean mixing lengths", add number of measurements

L. 294: delete "the" from "the both"

L. 334: replace "Niels-Bohr Institute" with "PICE" to stay consistent

Fig A1: Please refer to this detailed schematic in caption of Fig 1 in main text.

L. 463: replace "steam" with "stream"

**Bibliography:**

Calonne, N., Geindreau, C., Flin, F., Morin, S., Lesaffre, B., Rolland Du Roscoat, S., and Charrier, P.: 3-D image-based numerical computations of snow permeability: links to specific surface area, density, and microstructural anisotropy, The Cryosphere, 6, 939–951, https://doi.org/10.5194/tc-6-939-2012, 2012.

Colbeck, S. C.: The capillary effects on water percolation in homogeneous snow, J. Glaciol., 13, 85–97, https://doi.org/10.3189/S002214300002339X, 1974.

Jones, T. R., White, J. W. C., Steig, E. J., Vaughn, B. H., Morris, V., Gkinis, V., Markle, B. R., and Schoenemann, S. W.: Improved methodologies for continuous-flow analysis of stable water isotopes in ice cores, Atmospheric Measurement Techniques, 10, 617–632, https://doi.org/10.5194/amt-10-617-2017, 2017.

Yamaguchi, S., Katsushima, T., Sato, A., and Kumakura, T.: Water retention curve of snow with different grain sizes, Cold Regions Science and Technology, 64, 87–93, https://doi.org/10.1016/j.coldregions.2010.05.008, 2010.

---

## Author Response (AR1)

**Answers to Topical Editor**

Dear Authors,

The manuscript meets the criteria to be put forward into the interactive discussion and sent to referees in its current form. Nevertheless, and depending on the referees' comments, I recommend that authors consider strengthening the scientific discussion of their results during the revision phase, in order to mitigate the essentially technical nature of the message. I have also noted a few issues that would need clarification:

-Section 2.2 presents two approaches for estimating mixing lengths, but I have not found any further reference to the second approach in the rest of the manuscript.

Answer to the editor:

Thank you very much for the evaluation and naming the critical points of the manuscript.

In the revised version we a) strengthen the focus of our study by rephrasing the abstract and to a small extent the introduction. We re-named the subchapters of the method and result sections in order to better indicate the different approaches to estimate mixing length and their results.To this end we clarified that the approaches give similar results which is why we continued with the "normal distribution" in the manuscript. Finally, we added/reordered the discussion such that it contains a more in-depth discussion of the results and also helps a bit in clarifying the main purpose of this study.

Please find our (colored) answers to your comments and the reviewers comments below.

Changes in the text:

- Abstract: we rephrased the abstract to clarify the motivation and major outcome of this study
- Introduction: Lines 48ff: we added/rephrased the text in order to strengthen the motivation.
- Method and Result: We renamed/added titles for the subchapter to better address the two different approaches
- Section 2.2 now contains the following chapters: "Characterization of mixing using step functions" and "Characterization of mixing by comparison to discrete samples"
- Section 3 now contains the following subsections: "3.3 Mixing length derived from step function tests" and "3.4 Mixing length derived from comparison to discrete measurements"
- For the mixing length obtained by step functions tests, two approaches are used, to fit the data - the normal Gaussian distribution function and the skewed distribution function. As the results are similar we use the normal Gaussian distribution function in the rest of the manuscript. In order to clarify, we added a sentence at the end of section 2.2.: "Our results show very small differences in the resulting mixing lengths obtained by the different approaches (Table 2). We therefore focus this work on the straight forward Gaussian approach to determine the diffusion lengths and assessment of contribution to the overall mixing"
- Results: we modified table 2, such that it includes both results, from normal distribution function and skew distribution function.

-Line 169: I am not sure about the meaning of "a calotte of 25 mm thickness lengthwise".

Answer to the editor:

The term "calotte" refers to the longitudinal sub-section of a 1m long ice core. We agree to find another description.

Changes in the text:

- We rephrased the sentence to: "From all snow cores a longitudinal section (slice) of 25mm thickness was cut. This slice was further cut into discrete subsamples of 22 mm size in vertical resolution.

-Line 184: Where is it observed that "instrumental drift starts earlier than 104 s"?

Answer to the editor:

From our Allan deviation test, we observe that after a time between $10^3$ and $10^4$ seconds of integration time, the standard deviation increases (Figure 3), indicating the instrumental drift. We will clarify the link within the text. (But maybe this was a misreading due to formatting - saying 104 instead of $10^4$ seconds).

Change in the text:

- We corrected the format of the number to say $10^4$ seconds
- We rephrase this statement to:"As one run (melting 1-meter liner with CFA) takes up to 2000 seconds, our measurement time window of this one meter is well within the non-affected time window for drift. Therefore, one single calibration for each single meter is necessary."

How did the authors come to the conclusion that "a single calibration per core is necessary"?

Answer to the editor:

Based on our Allan deviation test we find that any measurements longer than $10^4$ are affected by instrumental drift. As stated in Jones et al, (2017), in principle a longer integration time leads to better precision until the drift of the instruments increases (due to instrumental factors, temperature etc..). This is what we observe between $10^3$ and $10^4$ seconds (for deuterium) and $>10^4$ (for d18O. Until then, any measurements are not affected by drift. As the measurement of one core (here referring to 1m liner) takes approx. than 2000 seconds, one calibration per meter (liner) is sufficient and needed. We will clarify this in the text.

Change in the text:

- We rephrase this statement to:"As one run (melting 1-meter liner with CFA) takes up to 2000 seconds, our measurement time window of this one meter is well within the non-affected time window for drift. Therefore, one single calibration for each single meter is necessary."

-Section 3.4: The phase offset between the continuous and discrete measurements is attributed to "short-term changes in the melt speed". Isn't it possible to have direct measurements of the melt speed by the laser sensor shown in Fig. 1?

Answer to the editor:

Indeed, the melt speed is directly measured continuously at a resolution 0.1seconds, then averaged to 1 second.

The depth assignment is in fact corrected for these density-induced variations in melt speed. So while we consider the main reason for the off-set being the problem of assigning discretely cutted samples vs continuous measurements, we wanted to include this issue as one part. But it is probably not the main source. We will clarify this in the text.

Changes in the text:

- We add: "One reason for the mismatch could be the short-term fluctuation in the melt speed induced by density variations. However, the melt speed is measured with high resolution and variations are included in the depth assignment. We therefore assume that the depth assignment from the CFA is quite accurate. On the other hand the depth assignment of the discrete samples is less accurate, as the subsampling generates uncertainty in the exact size and position of each sample (i.e. loss of material due to the cutting). This uncertainty in the depth assignment and the corresponding matching to the CFA data may be one major reason for the observed mismatch of discrete and continuous data."

I was also bothered by the heavy reliance on acronyms throughout the text. To facilitate reading, I would recommend reducing the number of acronyms.

Answer to editor:

Indeed, the large number of technical and mathematical acronyms is disturbing for the reader. For most of them, we suggest writing their original names, while keeping some for readability as they are often used.

Changes in text:

- CFA is kept for Continuous Flow Analysis.
- MH is replaced by melt head
- DB is replaced by debubbler
- PP is kept and defined as Peristaltic Pump
- BD-i are replaced by Bubble Detectors
- FM is replaced by Flow Meter
- PFA is replaced by Perfluoroalkox
- OD is replaced by outer diameter
- ID is replaced by inner diameter
- MQ is kept for MilliQ
- SSH is replaced by snow sample holder
- PS is replaced by laser positioning
- W is replaced by weight
- MF is replaced by manifold
- CC-i are kept for conductivity cells
- CRDS is kept for Cavity Ring Down Spectrometer
- CDF and PDF are kept, for cumulative distribution function and probability density function, respectively.
- The names of snow sample liners (KF13 - KF18) are kept.

We thank the reviewer for the in-depth discussion and helpful comments.

Please elaborate on the following topics:

- Snow property influences: Both water isotope diffusion during storage and percolation are influenced by snow properties such as density and grain size which should be discussed in both sections. Please also add a statement about the density of the snow cores analyzed in this study.

Answer to the reviewer - microstructure and density vs stable water isotopes:

In general the interplay of density and microstructure (grain size) in the snow originates from layering, i.e. the deposition of single layers. This structure is not seen / kept in the isotope signal, as diffusion smoothes very quickly. So we do not expect to see influences of layered variations of density and/or microstructure in the isotope signal. A in-depth study of high-resolution microstructure (including density) and its link to trace components and stable water isotopes at the Kohnen Station was given here: Moser et al., 2020, https://doi.org/10.3389/feart.2020.00023

The only potential effect is that strong changes in density may affect the melt speed - which is actually considered by the highly-resolved measurement of melt speed, and the amount of percolation, when for example changing from a lower-density (larger grains) layer to a higher-density layer (smaller grains). However, we consider this effect to be much smaller compared to the overall mixing. In fact we can show that changes in the melt speed (when comparing different CFA runs) do not affect the overall mixing.

Changes in the text:

- We add in the paragraph: Characterization of mixing length by comparison to discrete samples the following sentence: "High-resolution density measurements display the layered character of the snow, which induces mm-to-cm variations in density and microstructure. However, stable water isotopes do not capture these variations, as the diffusion on site smoothes the signal rather quickly (Moser et al. 2020)."
- We add the new paragraph 4.2 in the discussion, where the effect of variations in melt speed to the mixing length is discussed.
- We moved the paragraph on the storage effect to the discussion. We add the following sentence: "However the strength of the observed diffusion during storage is due to the low density of the snow cores and we do not expect such a strong change for firn and ice core samples from greater depth and with higher density."

- Melt rate: How was the optimal melt rate chosen and how does this influence the results? Isotope CFA systems are often run in parallel with other instruments and thus the overall CFA melt rate is constrained by more than the water isotope line alone, but if high-resolution water isotope records from snow were the primary goal, which melt rate should be chosen, and what are the constraints? Please also add the statistics of the melt rates of your experiments to the text, both intra and inter snow core variability. Currently, only the mean melt rates are given in the figure captions and are thus hard to find.

Answer to the reviewer:

While the major focus of the CFA experiments lay on the stable water isotope measurements, other lines were operated (see figure 1). Thus, the amount of connected lines determines the required melt water to be generated and accordingly the required flow - here approy. 8ml/minute. The estimate of the necessary melt speed also considers the diameter of the inner channel, being 30mm, and the density of the snow cores (on average 340kg.m-3). Based on these assumptions our

constraint of the melt speed was ~33 mm/min. However, in order to assess any dependency of the observed mixing to melt speed, we applied different melt speeds between 2.87 and 4.95 cm/min.

Changes in the text (discussion):

We added a section 4.2 in the discussion:

- Further we explored the sensitivity of mixing length to melt speed. We melted the snow cores at different melt speeds, varying from 28.7 mm per minute to 49.5 mm per minute (Table 4). As the obtained mixing lengths do not show any trend (see Table 3), we are confident that the mixing length is not sensitive to melt speed variations. We however observed variability of melt speed within each core between 10 and 20 % of the mean speed (Table 5), which can be related to small-scale variations of snow density or friction within the sample holder.

- We added a table displaying the different melt speeds and the standard deviation:

| | KF13 | KF14 | KF15 | KF16 | KF17 | KF18 |
|---|---|---|---|---|---|---|
| Average speed | x | 28.7 | 31 | 33 | 46 | 49.5 |
| Standard deviation | x | 3.8 (13%) | 4.4 (14%) | 5.6 (17%) | 9 (19%) | 5.3 (11%) |

**New Table 4:** Melt speed statistics for the different snow cores, expressed in mm.min$^{-1}$. The upper row shows the averaged melt speed of each run, while the lower row indicates the corresponding variability (standard deviation observed in mm.min-1), and in parenthesis as a percentage of the mean value.

 - In line 159 the authors declare that they will compare two different diffusion length calculations (normal CDF and two lognormal CDFS) but these results are not presented, although they would be of interest to the reader as the asymmetry in the impulse response is obvious from Figure 5b.

Answer to the reviewer:

As the results are similar we use the normal Gaussian distribution function in the rest of the manuscript. In order to clarify, we added a sentence at the end of section 2.2., but include both results in table 2

Changes in the text:

- We added a sentence at the end of section 2.2.: "Our results show very small differences in the resulting mixing lengths obtained by the different approaches (Table 2). We therefore focus this work on the straight forward Gaussian approach to determine the diffusion lengths and assessment of contributions to the overall mixing"

- We modified Table 2, such that it includes both results, from Gaussian distribution function and Skew distribution function.

|  | MH-filled | CRDS-line | MH-empty | MH | DB |
|---|---|---|---|---|---|
| Skew | *20.0 (1.9)* | *11.5 (2.2)* | *13.7 (1.1)* |  |  |
|  | **12,7 (1.2)** | **7.3 (1.4)** | **8.7 (0.7)** | **9.2** | **4.7** |
| Normal | *21.6 (2.4)* | *12.6 (1.8)* | *14.5 (1.2)* |  |  |
|  | **13.6 (1.5)** | **8.0 (1.1)** | **9.2 (0.7)** | **10.0** | **4.5** |

**Updated Table 2:** $\delta$D means mixing length derived from the skew and normal PDFs, expressed in seconds (italic) and in mm (bold). Values in parenthesis represent 1SD. The conversion of seconds to mm is based on a melt-speed of 38 mm.min-1. $\sigma_{MH}$ and $\sigma_{MH\ to\ WI\text{-}SV}$ correspond to differences in quadrature between $\sigma_{MH\text{-}filled}$ and $\sigma_{MH\text{-}empty}$, and between $\sigma_{MH\text{-}empty}$ and $\sigma_{CRDS\text{-}line}$, respectively.

- It would be interesting to see a deconvoluted CFA record in comparison to the discrete samples of 2019 as this would be the final post-processing step for a snow CFA campaign when producing data for climate analysis. Arguably, the reverse was done to find the percolation mixing length contribution, but these data are not visualized and the statistics (RMS, l. 267) on the agreement with the convoluted discrete samples and the CFA data are not given. As such the reader has no sense for the agreement between discrete and post-processed CFA data. A visualization of such a final data product would help the reader understand the importance of the system characterization and demonstrate the utility of the Snow CFA line.

Answer to the reviewer:

While we agree that this is of great interest, we emphasize that it is not the goal of this study to correct for diffusion. We however did such tests on a shallow firn core in the past. For the full convolution assessment and deconvolution (which includes an additional work on frequency filtering during the deconvolution step) we refer to a Master thesis (not published). If the reviewer is interested: Hannah Meyer (2020, Uni Bremen), "Characterization of a continuous-flow analysis facility and development of data correction techniques: An analysis of the instrument-induced mixing of the water isotopic record in firn cores".

In the study presented here we aim at assessing and quantifying the quality of the measurement.

Specific comments:

L. 21: The sentence "With our obtained mixing lengths…" is difficult to understand. Please rephrase.

The abstract has been rephrased in order to clarify the main focus of the study. It now reads: "We systematically determine the mixing length at different parts of the system. We measure and analyze six snow cores from Kohnen station, Antarctica, and find the largest contribution to mixing to originate in the percolation of melt water on top of the melt head."

L. 46: "percolation" is often used in snow science to describe the vertical, gravity-driven water flow in a snowpack. To avoid confusion, please consider exchanging the term "percolation" with "wicking" (as done in (Jones et al., 2017)) or including a short terminology explanation.

Answer to the reviewer:

We here use the term describing the upflow of meltwater into the snowpack due to capillary effects, i.e. the melt water gets sucked into the snow above.

In the literature dealing with this phenomenon at the CFA melt head, both terms are used, i.e. "wicking" in for example Erich C. Osterberg, Michael J. Handley, Sharon B. Sneed, Paul A. Mayewski, and Karl J. Kreutz Environmental Science & Technology **2006** *40* (10), 3355-3361, DOI: 10.1021/es052536w

but also the term "percolation" is used, for example in: Kjær, H. A., Lolk Hauge, L., Simonsen, M., Yoldi, Z., Koldtoft, I., Hörhold, M., Freitag, J., Kipfstuhl, S., Svensson, A., and Vallelonga, P.: A portable lightweight in situ analysis (LISA) box for ice and snow analysis, The Cryosphere, 15, 3719–3730, https://doi.org/10.5194/tc-15-3719-2021, 2021.

We here stick to the term "percolation", but add a sentence with a definition.

Changes in the text:

In the introduction, when the term "percolation" is used for the first time, we add/modify the sentences to:

"However, in the past, firn and ice cores were analyzed only starting in several meter depth. Due to the very high porosity and poor stability of surface snow, the upper meter of the polar snow pack has not been analyzed by CFA, but was occasionally sampled discreetly. One obstacle for measurements with CFA is the strong percolation taking place in the highly porous snow. We here refer to "percolation" as the upward movement of meltwater into the snow due to capillary effects. The percolation in the snow above the melt head leads to mixing of melt water of adjacent snow layers. This physical mixing inevitably leads to a smoothing of the derived snow record including the isotope signal (Gkinis et al., 2011)."

L. 46: The authors mention here that percolation is the reason why snow cores have not been measured routinely with CFA. The results presented in this paper support this statement. However, it is not made clear whether the authors conclude that their presented method has overcome this hurdle or which limitations remain with the system presented in this manuscript. I suggest the authors include a "best-practices" or "limitations" section in the discussion.

Answer to the reviewer:

In our study we quantify for the first time the effect of percolation on the overall mixing of the isotope signal, by separating this effect from the instrumental contribution.

We further suggest as a "best-practice", the development of a new melt head that can better handle the melt water, thereby minimizing the percolation. This paragraph has been moved to the discussion.

Changes in the text:

We moved/added a paragraph into the discussion "Suggestion to address the percolation"

L. 54: In the introduction it would be helpful to include a short paragraph on the characteristic "mixing of the system" or "smoothing" that all water CFA systems suffer from and that is the drawback of CFA analysis of ice cores. In the current version the "mixing of the system" is first mentioned in L.54 but it is not explained until Section 2.2, (L. 136). Since the "diffusion length" is the focus of the analyses in this manuscript it would help to explain these terms already in the introduction.

> Changes in the text:
>
> We added/modified the sentences in the introduction to:
>
> "However, in the past, firn and ice cores were only analyzed starting at several meter depth. Due to the very high porosity and poor stability of surface snow, the upper meter of the polar snow pack has not been analyzed by CFA, and was only occasionally sampled discretely. One challenge for CFA measurements is the strong percolation taking place in the highly porous snow. Here, we refer to "percolation" as the upward movement of meltwater into the snow due to capillary effects. Percolation in the snow above the melt head leads to mixing of melt water from adjacent snow layers. This physical mixing inevitably smooths the derived snow record including the isotope signal (Gkinis et al., 2011)."

L. 59: Is this set-up significantly different from other CFA systems that are being used in ice core analysis laboratories? Explain the differences or cite studies where this setup has been used previously.

> Answer to the reviewer:
>
> Indeed, the CFA system used here is similar to the system used for ice-cores. The specificity of our system refers to the adapted melting unit: the melt head for snow cores combined with the core guide, allowing to hold and melt snow cores of 10cm in diameter (where usual CFA applications use a melt head to melt a stick of 3.4 x3.4 cm size.
>
> Changes in the text:
>
> - In the introduction we rephrased the sentence to: "In order to enable stable water isotopes measurements in snow cores by CFA, we modified the melting unit of our CFA-system developed a (...)"
> - In 2.1. we rephrased the sentence to: "The system for analyzing 1-meter snow-cores consists of a melting unit adapted to the geometry of the snow cores, a degassing unit, a (…)"
> - We changed the title of the paragraph to: "Melting unit adapted for snow cores"

L. 126: What are typical lag times between the different lines?

> Answer to the reviewer:
>
> During the corresponding melting experiments, we observed averaged lags of 39.8s for the conductivity line, 40.6s for the Micro-particles line, 70s for the Fraction collection line, and 145s for the water isotopes line. These delays are significantly different as each of them is defined by each line length and by the flow applied from the manifold to each detection unit (Fig.1).
>
> Changes in the text: None, as not relevant for the paper

L. 154: Were all mixing times (in s) converted to mixing lengths (in cm) using the same melt-rate? How stable was the melt-rate of the system during the experiments considering density variability in the snow cores?

Answer to the reviewer:

All mixing times (in seconds) were converted to mixing length using the same melt speed - using an average value of 38 mm/minute. This enabled us to derive consistency between the different contributors (percolation, instrument). We do not find an effect of variable melt speed to the derived mixing length. We now strengthen this finding by adding a paragraph in the discussion.

Changes in the text:

We add a paragraph 4.2 "The effect of variation of melt speed on the mixing length".

L. 166: If available, please add age and density of the analyzed snow cores.

Changes in the text:

In the paragraph "Characterization of mixing by comparison to discrete samples" we add the sentence: "The 1 meter average density of these snow cores ranges between 340 - 345 kg m-3 (Münch et al., 2017)."

L. 178: Were the Allan variance tests performed by injecting MQ water at the Master IV valve? Would you expect the stability to be different for MQ water injected at the MH?

Answer to the reviewer:

The Allan Variance test was performed by injecting MQ at the WI selection valve. This is needed in order to get a stable stream. If the MQ was added as the Master IV (or further upstream, i.e. MH), the stream stability could potentially be affected, leading to a bias in our check of the analytical Picarro performances.

Changes in the text:

We modified the sentence to: "(...) where over a period of more than 12 hours MQ water is continuously injected at the selection valve for water isotopes (Fig.1)."

Fig. 3: I suggest removing the upper row of the figure since both plots don't add much information

Changes in the text/figure: The upper row of figure 3 has been removed.

Fig 4: Consider moving this figure to the Appendix.

Changes in the text: Previous Figure 4 was moved to Appendix B, now being Figure B1

L. 207: Is the diffusion length dependent on the step size, i.e. the standards used? Please add which standards were used to simulate the step.

Answer to the reviewer:

The isotopic steps were realized during each calibration procedure. Each of these procedures consists of a switch between MQ to standard NZE (~100 permille drop in dD), NZE to TD1(~120 permille drop), and TD1 to JASE (~140 permille). Based on our 15 steps applied, we did not observe relationships between isotopic steps and diffusion times/lengths. Regarding the isotopic steps applied at the melt head (Filled or empty), the switch was realized between the standard TD1 and MQ water (>200 permille step).

Changes in the text:

We rephrase this section to:

"We estimate the mixing length using isotopic liquid standards across different experiments (details given in Table C1 in the supplement). First, during each calibration procedure, three abrupt isotopic changes are applied (dataset CRDS-line, Fig. 4) at the water isotopes selection valve (Fig.1) . These steps range between ~100 ‰ and ~140 ‰ (Appendix C), and show no dependency between isotopic step size and diffusion length. In a second experiment, we applied an isotopic step (~230 ‰) at the melt head with its concave volume filled (dataset MH-filled, Fig.5). Finally, the same step was then applied with the concave volume empty (dataset MH-empty, Fig.5)."

L. 208: I would appreciate a table listing the different experiments and the corresponding naming conventions of the different s to ease the reading

Changes in the text

In order to clarify all the experiments realized, we added a summarizing Table in supplement. As shown above we refer to this additional Table in the text.

| Experiment | Abbreviation | Number of runs | Standards switch | Isotopic step size |
|---|---|---|---|---|
| Water Isotopes Calibration (step 1) | CRDS-line | 5 | MQ to NZE | ~100 ‰ |
| Water Isotopes Calibration (step 2) | CRDS-line | 5 | NZE to TD1 | ~120 ‰ |
| Water Isotopes Calibration (step 3) | CRDS-line | 5 | TD1 to JASE | ~140 ‰ |
| Melt head filled | MH-filled | 4 | TD1 to MQ | ~ 230 ‰ |
| Melt head empty | MH_empty | 4 | TD1 to MQ | ~ 230 ‰ |

**Table C1**: Details of the different experiments conducted to assess the overall instrumental mixing, and separating the different contributions along the setup.

L. 217: Please also give the results for d18O, even if you focus the discussion on dD

Answer to the reviewer:

We agree that presenting all results is informative. However, to avoid potential confusion from including all numbers in a single table, we modified Table 2 to include all results for dD and added an additional table in the appendix with the results for d18O.

Changes in the text:

We rephrase the sentence to: "The results for both isotopologues $\delta D$ and $\delta^{18}O$ are very similar (Appendix C, Table C2), consistently showing a slightly longer diffusion length for $\delta D$. Therefore, we focus our study on the results for $\delta D$ (Table 2)."

L. 249: Can this depth assignment mismatch be a result of the system lag between the core hitting the melt head and the time the CRDS is measuring the respective sample? Or is this lag time accounted for?

Answer to the reviewer:

This lag between the snow hitting the melt head and the melt water arriving at the CRDS is accounted for (as for every other analytical line in the system). We also discuss other factors such as density variations (not affecting the depth assignment) and the cutting of discrete samples.

Changes in the text:

- We add: "One reason for the mismatch could be the short-term fluctuation in the melt speed induced by density variations. However, the melt speed is measured with high resolution and variations are included in the depth assignment. We therefore assume that the depth assignment from the CFA is quite accurate. On the other hand the depth assignment of the discrete samples is less accurate, as the subsampling generates uncertainty in the exact size and position of each sample (i.e. loss of material due to the cutting). This uncertainty in the depth assignment and the corresponding matching to the CFA data may be one major reason for the observed mismatch of discrete and continuous data."

L. 267: How well do the convoluted and continuous records agree? Please give RMS values.

Answer to the reviewer:

It's an automated routine finding the best match and providing the respective mixing length. So we do not have the specific numbers, but from previous experiments, assuming a comparable behavior, we believe RMS is in the range of 0.1‰ for d18O as shown in the Table below (from Hannah Meyer's Master Thesis).

117

**Appendix D**

**$\sigma_{CFA}$ for the depth domain approach-Additional tables**

TABLE D.1: $\sigma_{CFA}$ for $G_{normal}$ and $G_{skew}$ retrieved from the depth domain approach over ~ 4 $m$ length and the mean RMS distances of CFA and discrete before and after the convolution for $\delta^{18}O$.

| meters | $\sigma_{normal}$ [cm] | CFA-discrete convolved with $\sigma_{normal}$ [‰] | $\sigma_{skew}$ [cm] | skewness factor $\alpha$ | CFA-discrete convolved with $\sigma_{skew}$ [‰] | CFA-discrete [‰] |
|---|---|---|---|---|---|---|
| 5-9 | 1.49 | 0.14 | 1.46 | 9.85 | 0.14 | 0.15 |
| 9-14 | 1.87 | 0.11 | 1.87 | 1.61 | 0.11 | 0.14 |
| 14-18.01 | 1.72 | 0.14 | 1.63 | 1.07 | 0.09 | 0.15 |
| 18.01-21.6 | 1.91 | 0.13 | 1.76 | 0.86 | 0.08 | 0.14 |
| 29-32.16 | 1.15 | 0.10 | 2.00 | 12.67 | 0.1 | 0.10 |
| 5-14 | 1.73 | 0.13 | 1.71 | 2.00 | 0.13 | 0.15 |
| 14-21.6 | 1.89 | 0.13 | 1.78 | 0.90 | 0.09 | 0.15 |
| 5-21.6 | 1.78 | 0.14 | 1.75 | 1.55 | 0.12 | 0.15 |

Table from the Master Thesis by Hanna Meyer

L. 280: Please give variability of this percolation diffusion length. Discuss the dependence on melt-rate or snow properties (e.g. (Calonne et al., 2012; Yamaguchi et al., 2010; Colbeck, 1974)?

Answer to the reviewer:

If you are referring to mixing length, we demonstrate that it is not dependent on melt speed (i.e. melt rate) or snow properties at this length-scale (1 m snow cores).

Changes in the text:

Please see our reply and related changes in the text above (first comment of the reviewer & the comment on melt rate)

L. 285: Please add short statement on how the snow cores were stored (temperature, sealing, …)

Changes in the text:

We added: "The snow cores were stored in carbon tubes, sealed at each end with plastic bags (WhirlPack), and kept inside a Styrofoam box at -25°C."

Table 4: Please add the differences between the two measurement campaigns to the table

Answer to the reviewer:

There have been no differences in the campaigns, just the timing

L. 337: The effect of density or other snow properties on the percolation strength is not discussed up to this point. Please include in discussion.

Answer to the reviewer:

Please see our answer and related changes above (in the first comment)

L. 339: As mentioned above, a clear conclusion and recommendation is missing whether the presented snow-CFA system is recommendable to use and, if not, what restrictions or limitations (accumulation threshold, density etc) apply.

Answer to the reviewer:

In our study, we quantify for the first time the effect of percolation on the overall mixing of the isotope signal, by separating this effect from the instrumental contribution.

We further suggest as "best-practice" to develop a new melt head that is better able to deal with the melt water , thereby minimizing the percolation. This paragraph has been moved to the discussion.

Changes in the text:

We moved/added a paragraph into the discussion "Suggestion to address the percolation"

Appendix B: Please add short introduction to the two presented figures and cite the model that is being used. I recommend highlighting AWI storage temperature and Kohnen annual mean temperature to Fig B2.

Changes in the text:

- We added to Figure D1: "Exemplary firn-diffusion length estimates for firn samples with a density of 370 kg.m⁻³ as function of storage time. The firn diffusion length is computed

based on the model by Gkinis et al. (2014), using a temperature of -20⁰C and an
accumulation rate of 75 mm"

- We updated Figure D2:

[Figure]

**Figure D2:** Diffusivity coefficients versus temperature, constraints as above, and displaying the
annual mean temperature at Kohnen station (-43⁰C, Weinhart et al., 2020) and the storage
temperature (-20⁰C).

Technical corrections:

L. 21: What does "continuous analyze" mean? → removed (Abstract rephrased)

L. 28: delete one "stable" → removed (Abstract rephrased)

L. 29: Capitalize "East Antarctic Ice Sheet" → DONE

L. 31: change "variabilities" to "variability" → DONE

L. 52: Delete "In order" → we kept this

L. 56: replace "signal, with…" with "signal, and show that…" → Rephrased to: "(...) isotope signal, and
show that the percolation above the melt head is the major contributor."

Fig. 1: What does "PP" stand for? → Peristaltic Pump, explanation has been added to the figure caption

L. 80: melting → renamed to "Melting unit adapted for snow cores"

L. 143: Refer here to Fig. 5 → DONE

L. 183: replace "optimal" with "minimal" → DONE

Fig. 3 and other Figures: Please add panel labels to all plots and refer to them in the captions

- Figure 1 - no panels
- Figure 2 - panels are labeled with a,b, c
- Figure 3 - only 2 subpanels left, referred to as "left" and "right" panel
- Figure 4 - (previous figure 5) - figure caption is changed to refer to the "upper" and "lower" panel

- Figure 5 (previous figure 6) - figure captions is modified such, that it refers to "upper" and "lower" panel
- Figure 6 (previous figure 7) - no panels

Table 1: Please add uncertainties of these in-house standards. → DONE, see table1

Table 2 caption: Change "means mixing length" to "mean mixing lengths", add number of measurements → figure caption changed, number of measurements is given in table C1

L. 294: delete "the" from "the both" → DONE

L. 334: replace "Niels-Bohr Institute" with "PICE" to stay consistent → deleted "Niels-Bohr Institute

Fig A1: Please refer to this detailed schematic in the caption of Fig 1 in main text. → DONE

This manuscript focuses on the continuous measurements of water isotopes in firn using a continuous flow analytical (CFA) set-up and uses this set-up, combined with disctrete analyses, to address the effect of diffusion of water isotopes during storage. The study presents some new measurements from 1 m sections (2.4 to 3.4 m) on 6 firn cores. Because CFA is affected to some mixing due to percolation, analyses of discrete samples performed in 2015 and 2019 over the same sections are also used to infer the effect of diffusion over 4 years. The authors deduce a diffusion length of 45 mm for storage diffusion during 4 years.

The manuscript is short, useful and in general well written. I suggest to accept it but I would like to suggest a few points of the study which should be explained in more details before acceptance.

> We thank the reviewer for the helpful comments.

- It was not clear to me what was new in the set-up presented here compared to the previous one in addition to the change of the melting head. It could be better explained.

  > Answer to the reviewer:

  > Indeed, the CFA system used here is similar to the system used for ice-cores. The specificity of our system refers to the adapted melting unit: the melt head for snow cores combined with the core guide, allowing to hold and melt snow cores of 10cm in diameter (where usual CFA applications use a melt head to melt a stick of 3.4 x3.4 cm size.

  > Changes in the text:

  > - In the introduction we rephrased the sentence to: "In order to enable stable water isotopes measurements in snow cores by CFA, we modified the melting unit of our CFA-system developed a (...)"
  > - In 2.1. we rephrased the sentence to: "The system to analyze 1-meter snow-cores consists of a melting unit adapted to the snow cores geometry, a degassing unit, a (…)"
  > - We changed the title of the paragraph to: "Melting unit adapted for snow cores"

- Is there any comparison between the performances of the old and the new melt-head for firn analysis and how could the improvement be quantified?

  > Answer to the reviewer: Usually the melt heads for firn and ice are designed to follow the geometry of the ice sample, i.e. for most ice core projects a square of 3.4 x 3.4 cm,which is cut from the ice core sample prior to the measurement campaign. The melt head used here (and in Kjaer et al…) has a different geometry - as the snow cannot be subsampled, the full core is melted. The melt head has the geometry of a 10cm diameter snow core. Therefore, a comparison of the performance of the two meld heads with respect to snow is difficult.

  > Suggested change in the text: None

- Why was only the section between 2.4 and 3.4 m studied ? Why not studying the effect of diffusion during storage at different densities (e.g. a section of very low density on the section covering 1 m at the very top of the firn, a section of 1 m at 3 m depth, a section of 1 m at 10 m depth and a section of 1 m near the close-off) ?

  Answer to the reviewer:

  Indeed, investigating the storage effect on samples with different density would be helpful. However, the storage effect was not the purpose of this study, but rather a side observation. We chose this depth interval for convenience.

  Changes in the text:

  We moved the paragraph to the discussion and rephrase it to:

  **4.4 Diffusion of the isotope signal during storage of snow cores**
  From comparing the CFA data with discrete measurements (Fig. 5), we observed a significant difference between the stable water isotopic profiles of the same snow cores sampled at different times.
  This difference is not related to instrumental induced mixing, but instead indicates the effect of long-term storage of snow samples. We assume that diffusion, known to occur in snow and firn (Gkinis et al., 2014) also occurs in the cold storage environment. Using our results, we can now quantity this storage-induced diffusion. Assuming that the mixing of the discrete-15 dataset corresponds to the mixing of the discrete-19 dataset convolved with an independent smoothing filter induced by the storage, comes: [TL1] [RD2] (Equation 8)

  Using the calculated mean mixing lengths $\sigma_{CFA\text{-}discrete19} = 30$ mm and $\sigma_{CFA\text{-}discrete15} = 54$ mm (Table 3), we estimate a diffusion length of approximately 45 mm. These findings indicate that during the 4 years of storage (from the first analysis in 2015 to the second analysis in 2019), the isotope signal in the snow cores was smoothed by this diffusion length.
  Additionally, we computed for each 1-meter long snow core the mean and variability (standard deviation) of the both discrete datasets (Table 5). The decrease in amplitude indicates an average attenuation of 0.54 ‰ and 4.5 ‰ for $\delta^{18}O$ and $\delta D$, respectively. The mean values show on average an enrichment of isotopic composition of +0.31 ‰ for $\delta^{18}O$ and +1.6 ‰ for $\delta D$, likely due to the repeated contact with laboratory-air when bags are opened, and the loss of sample (frost) in the bag.

| | $\delta^{18}O$ | $\delta^{18}O$ | $\delta D$ | $\delta D$ |
|---|---|---|---|---|
| | Discrete-15 | Discrete-19 | Discrete-15 | Discrete-19 |
| KF13 | -45.29 (1.3) | -45.076 (0.85) | -356.31 (11.93) | -355.77 (8.05) |
| KF14 | -45.14 (1.45) | -44.71 (0.85) | -354.85 (12.24) | -353.12 (7.69) |
| KF15 | -46.38 (2.39) | -45.91 (2.05) | -364.36 (19.53) | -361.51 (16.75) |
| KF16 | -44.95 (1.70) | -44.76 (1.16) | -353.87 (14.72) | -353.30 (10.23) |

**Table 5:** $\delta^{18}O$, $\delta D$ means (standard deviation) for each snow core discrete dataset, expressed in ‰.

We show the effect of storage on diffusion lengths for both isotopologues (Fig. D1 Appendix D) based on firn-diffusion model (Gkinis et al., 2014). The model run, assuming a "storage" temperature of -20⁰C, a density of 370 kg.m-3, an accumulation rate of 75 mm w.e. yr-1 (even though there is no accumulation during storage). For a time window of 4 years, we found a diffusion length for δD similar to our observations (Fig. D1). As the diffusivity coefficients are positively correlated to temperature, the diffusion during storage is likely of stronger magnitude than on the East-Antarctic plateau (Fig. D2, Appendix D). The strength of the observed diffusion during storage is due to the low density of the snow cores and we do not expect such a strong change for firn and ice core samples from greater depths, where the density is higher.

- Is it possible to make some recommendations from this study on the storage conditions ? For example, depending on the accumulation rate at each site, the diffusion may affect the recording of the seasonal signal. Is it possible to say that below a certain accumulation rate, the seasonal signal is no more visible after « a certain number » (to be precised) of years of storage at -20°C ?

  Answer to the reviewer:

  We expect the diffusion during storage to be strongest in snow cores, but likely not critical for high-density firn and ice cores. However, further investigation is needed. Note, that at low accumulation rates, a "seasonal" signal is generally not visible in stable water isotopes due to diffusion - see Laepple et al. 2018: Laepple, T., Münch, T., Casado, M., Hoerhold, M., Landais, A., and Kipfstuhl, S.: On the similarity and apparent cycles of isotopic variations in East Antarctic snow pits, The Cryosphere, 12, 169–187, https://doi.org/10.5194/tc-12-169-2018, 2018.

Then, I have some minor remarks :

- l. 34-37 : can you precise at which timescales the ice cores of the East Antarctic plateau are dominated by noise ? As such, without any quantitative indication, these sentences are not very useful.

  Answer to the reviewer:

  There a different time scales involved, depending on the processes generating noise, We refer to the study by Casado et al. 2020: Casado, M., Münch, T., and Laepple, T.: Climatic information archived in ice cores: impact of intermittency and diffusion on the recorded isotopic signal in Antarctica, Clim. Past, 16, 1581–1598, https://doi.org/10.5194/cp-16-1581-2020, 2020.

  As for this study only short time scales are relevant; we restrict the introduction to stratigraphic noise, as done in line 34ff.

- Figure 4 : to see the (small) deviations of each data point from the regression line, it could be nice to show on top of each figure the difference in d18O for each standards between the real value and the value calculated from the measured value and the regression line.

  Answer to the reviewer:

  The figure displays the calibration line, i.e. the x-axis shows the measured value, the y-axis the defined one. The off-set between the measured and defined value (prior to calibration) can be visually taken from the figure.

Changes in the text:

- We now add the 1:1 line (which is similar to "calibrated" measured value), displaying the off-set BEFORE calibration.
- According to a comment by the other reviewer, we move this figure to the appendix.
- We apply the regression to each raw measurement and compare the calibrated value with the defined value. The results are displayed in table B1 in Appendix B:

| | Value after calibration | | Difference with defined value | |
|------|------------|-----------|------------|-----------|
| | $\delta^{18}O$ | $\delta D$ | $\delta^{18}O$ | $\delta D$ |
| NZE | -19.804 | -152.422 | -0.045 | -0.278 |
| TD1 | -33.935 | -266.732 | 0.084 | 0.532 |
| JASE | -50.181 | -392.247 | -0.039 | -0.252 |

**Table B1:** Deviations between calibrated and defined values. All values are expressed in ‰.

- Figure 6 : Is it possible to display an envelop showing the spread of the CFA profiles in addition to the stack ?

Answer to the reviewer:

Very good idea, It pictures the spread of the CFA profiles, which in fact displays the spatial variability.

Changes in the text:

We modified the figure accordingly

[Figure]

Lower panel: Mean datasets of the stacked CFA (black) and discrete (dotted) profiles of the snow cores KF-14, -15, -16. The gray shaded area displays the spread of the three CFA profiles.

- In general, I was wondering why you chose a value of 22 mm for the discrete sampling. It is probably not very convenient for the sampling of 1 m core section. Why not 20 mm ? What additionnal information (if any) could we learn with a higher resolution discrete sampling ?

Answer to the reviewer:

In fact, the intention was 20mm samples but including the thickness of the blade, when cuttin the samples, the net sampling resolution is 22 mm = 20mm sample thickness + 1mm material loss from the blade at each side of the sample = 22mm. This is usually not considered, but as we compare the discrete data to the CFA we refer to the "real" resolution.

We would not learn more /gain more information with respect to stable water isotopes using a higher resolution, as the isotope profile is smoothed. Infact, a higher resolution would increase the error due to depth assignment and the mass loss through cutting.

Changes in the text: None

- l. 319 : « Vostok » and not « Vostock » → removed

- I did not get exactly what should be improved on l. 324. Is it possible to further explain ?

Changes in the text (discussion, section 4.1: Suggestion to address the percolation):

We modified the statement to: "The significant contribution of the Picarro analyzer (CRDS-line, section 3.3) could be addressed as well, and requires likely a collaboration with the manufacturer to improve the analytical unit itself (e.g. reduction of the large volume at its inlet before a pressure-drop to the 40 Torr cavity)."